# Loss of Rnf31 and Vps4b sensitizes pancreatic cancer to T cell-mediated killing

Nina Frey[1,2], Luigi Tortola [1], David Egli[1], Sharan Janjuha[2], Tanja Rothgangl[2], Kim Fabiano Marquart[1,2], Franziska Ampenberger[1], Manfred Kopf[1] & Gerald Schwank [1,2 ✉]

Pancreatic ductal adenocarcinoma (PDA) is an inherently immune cell deprived tumor, characterized by desmoplastic stroma and suppressive immune cells. Here we systematically dissect PDA intrinsic mechanisms of immune evasion by in vitro and in vivo CRISPR screening, and identify Vps4b and Rnf31 as essential factors required for escaping CD8+ T cell killing. For *Vps4b* we find that inactivation impairs autophagy, resulting in increased accumulation of CD8+ T cell-derived granzyme B and subsequent tumor cell lysis. For *Rnf31* we demonstrate that it protects tumor cells from TNF-mediated caspase 8 cleavage and subsequent apoptosis induction, a mechanism that is conserved in human PDA organoids. Orthotopic transplantation of Vps4b- or Rnf31 deficient pancreatic tumors into immune competent mice, moreover, reveals increased CD8+ T cell infiltration and effector function, and markedly reduced tumor growth. Our work uncovers vulnerabilities in PDA that might be exploited to render these tumors more susceptible to the immune system.

[1] Institute of Molecular Health Sciences, ETH Zurich, Zurich, Switzerland. [2] Institute of Pharmacology and Toxicology, University of Zurich, Zurich, Switzerland.
✉email: gerald.schwank@uzh.ch

mmune evasion is a common trait of most human cancers. Through phenotypic changes, tumor cells evade recognition of effector T cells and modulate the tumor microenvironment to establish an immune-suppressive niche[1,2]. While immune checkpoint inhibition shows great potential for curative cancer treatment, pancreatic ductal adenocarcinoma (PDA) is largely refractory to immunotherapy[3,4]. Among the described mechanisms responsible for the highly effective immune evasion of PDA are (i) insufficient antigenicity[5], (ii) high expression of PD-L1[6], (iii) exclusion of dendritic cells while attracting T regulatory cells[2] and suppressive myeloid populations[7,8], and (iv) the sequestration of major histocompatibility complex class 1 (MHC-I)[9]. In order to better understand cell-autonomous mechanisms that protect tumors from immune clearance, genome-wide CRISPR-Cas9 screens have been performed in melanoma, renal-, colorectal- and breast cancer cell lines. Together they identified the interferon-γ (IFNγ) response, TNF-mediated NFκB signaling, and autophagy as core pathways involved in immune evasion across different cancer types[10–18]. However, to our knowledge, a comprehensive genetic analysis of anti-tumor immunity in PDA is missing, and it remains unclear which of the identified immune regulators are conserved for PDA or whether there are also genetic factors that are specific to PDA.

Here we use genome-wide in vitro CRISPR screening and targeted in vivo CRISPR screening to systematically reveal positive and negative regulators of cytotoxic T lymphocyte (CTL) sensitivity in PDA. In addition to previously described genes involved in the regulation CTL-mediated tumor cell killing, we identify *Rnf31* and *Vps4b* as central components for PDA immune escape in vitro and in vivo. *Rnf31* and *Vps4b* are classified as the strongest sensitizers to T cell killing in our screen. Results of this study suggest that *Rnf31*, as part of the linear ubiquitination chain assembly complex (LUBAC), mediates immune-escape by stabilizing anti-apoptotic proteins in the TNF pathway, and that *Vps4b*, as part of the autophagy machinery, reduces susceptibility to T cell-mediated tumor cell lysis by lowering intracellular granzyme B contents. Using an orthotopic transplantation model, we further demonstrate that loss of *Rnf31* and *Vps4b* gene function leads to reduced tumor growth and increasing T cell infiltration and effector function in vivo. The elucidated mechanisms of immune evasion in PDA provide potential strategies for enhancing the efficacy of cancer immunotherapies.

## Results

**A genome-wide CRISPR screen identifies regulators of immune evasion in PDA.** To identify genes modulating CTL-mediated killing of PDA we performed a pooled, genome-wide CRISPR knock-out screen in pancreatic cancer cells. We first engineered a PDA cell line derived from the autochthonous KPC mouse model (*Kras*^G12D, *Trp53*^R172H/+, *Pdx*-**C**re), which stably expresses *Sp*Cas9 and chicken ovalbumin (OVA). Cells were subsequently transduced with a murine single-guide (sg)RNA library targeting 19,647 genes at a 500× coverage[19]. To mimic cytotoxic T cell killing, we co-cultured cancer cells for three days with activated, OVA-specific CD8+ T cells (OT-I T cells) at a killing efficiency of approximately 70%, followed by a three-day recovery period prior to DNA isolation for analysis by next-generation sequencing (NGS) from the surviving cell population (Fig. 1a). Validating screening conditions and sufficient library representation, we observed a strong overlap of depleted sgRNAs targeting essential genes in OT-I T cell treated- and untreated KPC cells (Supplementary Fig. 1a, b). Next, we inspected differentially distributed sgRNAs, and defined genes targeted by enriched sgRNAs as resistors and genes targeted by depleted sgRNAs

as sensitizers for CTL-mediated killing (FDR < 0.1). We identified several genes with a well-characterized role in CTL-mediated killing in different cancer types, demonstrating that the previously described core cancer intrinsic CTL evasion gene network is also conserved in PDA (Fig. 1b, c)[10–12,14,20]. For example, genes associated with the IFNγ pathway (*Jak1, Jak2, Ifngr1, Ifngr2, Stat1*) and antigen presentation machinery (*B2m, Tap1*) conferred resistance to CTL-mediated PDA killing upon inactivation (Fig. 1c, d, Supplementary Fig. 1c), and genes regulating TNF-triggered apoptosis (*Cflar, Traf2*), NFκB signaling (*Nfkbia, Tnfaip3*) and autophagy (*Atg5, Atg7, Atg10, Atg12, Gabarapl2*) sensitized PDA cells to CTL-mediated killing upon inactivation (Fig. 1c, d, Supplementary Fig. 1c). Nevertheless, we also identified genes that have previously not been characterized as modulators of the CTL response, including the two strongest sensitizers *Rnf31* and *Vps4b*.

**A targeted CRISPR screen validates immune modulators in vivo.** To explore whether top candidates from the in vitro screen also affect CTL-mediated PDA killing in vivo we next performed a targeted library screen in mice. We generated a secondary library targeting 63 genes (hits with FDR < 0.1) with ten sgRNAs per gene, and containing 600 non-targeting control sgRNAs as well as seven positive control sgRNAs targeting ovalbumin. We furthermore omitted several IFNγ pathway components to avoid redundancy. The library was transduced into KPC-Cas9-OVA cells, which we subsequently orthotopically transplanted into pancreata of RAG1−/− mice. After tumor formation, we adoptively transferred activated CD8+ OT-I T cells to tumor-bearing mice and collected the residual tumors five days later (Fig. 2a). In vivo validated candidates showed consistent phenotypes across all mice (Fig. 2b), and in line with previous studies, we observed a substantial, albeit not complete overlap between in vitro and in vivo screening results (Fig. 2c, Supplementary Fig. 2a)[21]. Resistors of CTL evasion included well-known immune evasion genes, such as *Stat1* and *Casp8*, as well as the positive control *ovalbumin* (Fig. 2b). Among sensitizers of CTL killing—which are of particular therapeutic interest as they bear the potential to enhance anti-tumor immunity in PDA upon inhibition—were the previously described genes *Adar* and *Cflar*[22–24], as well as *Vps4b* and *Rnf31* (Fig. 2b, d). Since mechanistic insights of how *Vps4b* and *Rnf31* could regulate CTL sensitivity are still missing, and as both genes are transcriptionally upregulated in human PDAC samples compared to normal pancreatic tissue (Supplementary Fig. 2b), we decided to explore their role in PDA immune evasion more closely.

**A competition assay confirms the role of *Rnf31* and *Vps4b* in immune evasion.** We then performed arrayed validation of *Vps4b*- and *Rnf31*-mediated PDA sensitization to CTL killing in a competition assay. KPC-Cas9-OVA cells that express mCherry and carry *Vps4b*- or *Rnf31*- targeting sgRNAs were mixed in a 1:1 ratio with KPC-Cas9-OVA cells that express GFP and a non-targeting sgRNA. The formation of insertion- and deletion- (indel) mutations in the targeted loci was confirmed by deep sequencing (Supplementary Fig. 3a), and no changes in cell proliferation were observed upon gene disruption (Supplementary Fig. 3c). Mixtures of targeted and non-targeted KPC cells were co-cultured with CD8+ OT-I T cells and proportions of mCherry+ and GFP+ cells were determined by flow cytometry (Fig. 3a). As expected, unmutated mCherry+ KPC cells and control sgRNA treated GFP+ KPC cells grew equally well in the presence of OT-I T cells (Fig. 3b, c), and targeting *Stat1* shifted the ratio towards mCherry+ cells due to a defective IFNγ response (Fig. 3b, c, Supplementary Fig. 3g). In contrast, KPC cell lines mutant for *Vps4b* or *Rnf31* displayed a strong growth disadvantage under immune attack, leading to an

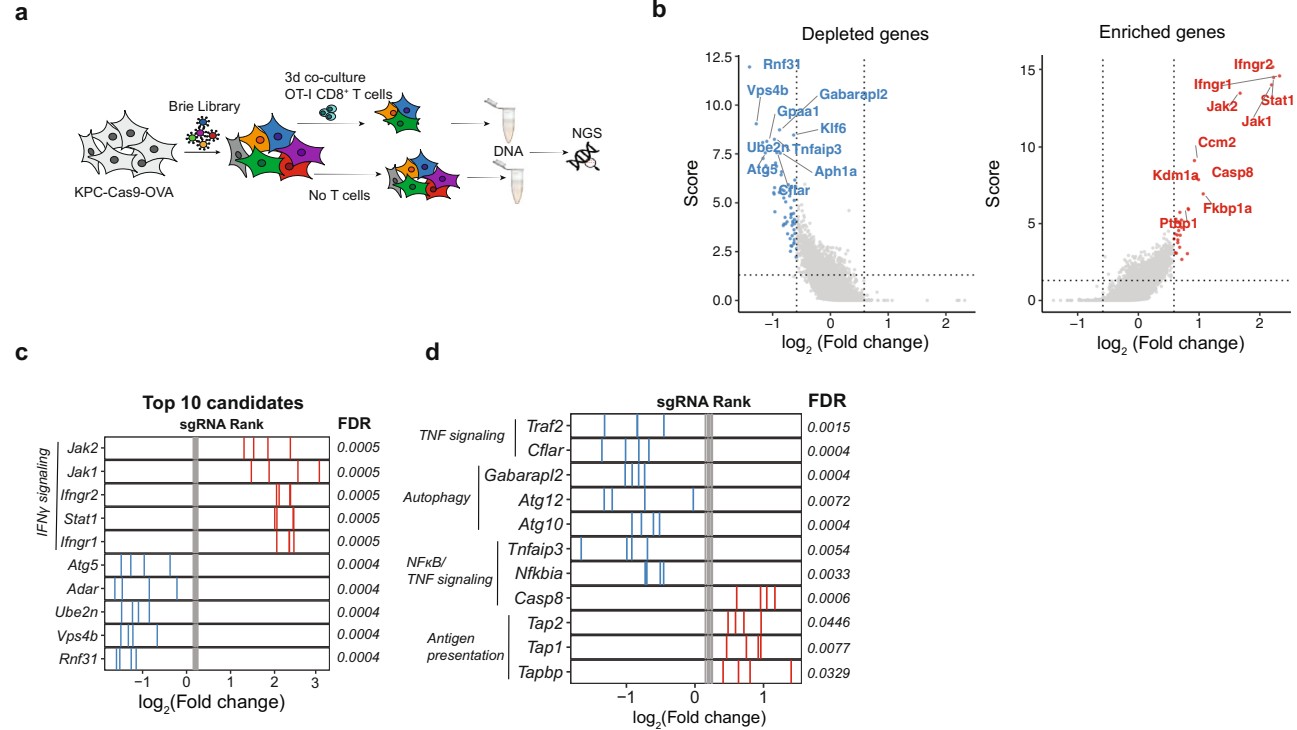

**Fig. 1 Genome-wide CRISPR screen in PDA cells reveals immune evasion mechanisms in vitro. a** Schematic of genome-wide in vitro CRISPR screen. **b** Volcano plot of top ten depleted (blue) and enriched (red) genes. Screening analysis was performed with MaGeCK RRA. **c** sgRANK of the top (red) and bottom (blue) five depleted genes are represented. Gray bars represent non-targeting sgRNAs. **d** sgRANK of the enriched (red) and depleted (blue) genes of different immune evasion pathways. Gray bars represent non-targeting sgRNAs. MaGeCK RRA analysis can be found in Supplementary Table 1. The genome-wide CRISPR screening data shown were derived from three independent biological replicates ($n = 3$).

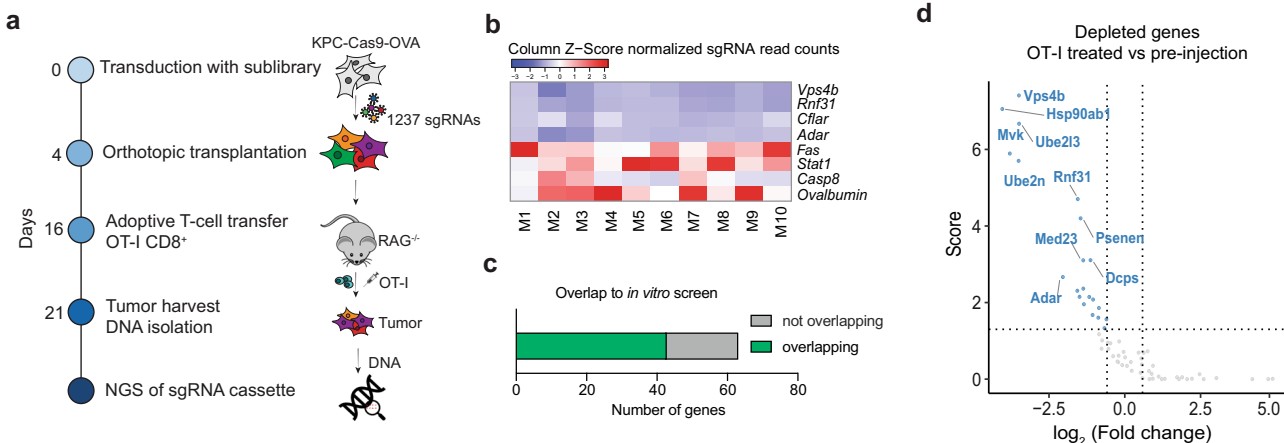

**Fig. 2 A targeted CRISPR library screen validates candidates in vivo. a** Schematic of the secondary CRISPR screen in vivo. **b** Heatmap of normalized read counts of sgRNAs across ten individual mice (M1–M10). **c** Bar diagram of sublibrary genes in comparison to their predicted phenotype (from in vitro screen). **d** Volcano plot of depleted (blue dots) genes of the sublibrary in vivo screen when comparing the pool of preinjected KPC cells to OT-I CD8+ T cell treated tumors. Sublibrary screening data were derived from a total of ten individual tumors (transplanted mice $n = 10$), transplanted in two experiments. Source data are provided as a Source Data file.

outgrowth of GFP+ control cells (Fig. 3b–d). Analyzing *Vps4b* and *Rnf31* in melanoma and breast cancer cell lines, moreover, indicate a partial conservation of this phenotype in other cancer entities, with loss of *Rnf31* also sensitizing B16 melanoma cells to CTL killing and loss of *Vps4b* leading to a trend towards higher sensitivity in EO771 breast cancer cells (Fig. 3d). Since Vps4b is part of endosomal sorting complex III (ESCRT-III), we also tested whether other members of the complex cause a similar phenotype. However, in line with our observation that none of these genes were identified

as a hit in our genome-wide CRISPR screen (Supplementary Table 2), arrayed elimination of *Vps4a*, *Vta1*, and *Chmp4b* did not sensitize KPC cells to CTL-mediated killing (Supplementary Fig. 3e).

**Functional characterization of the role of *Rnf31* and *Vps4b* in immune evasion.** Antigen presentation by major histocompatibility complex I (MHC-I) proteins is necessary for efficient antitumor immunity. As shown in a recent study, pancreatic cancer

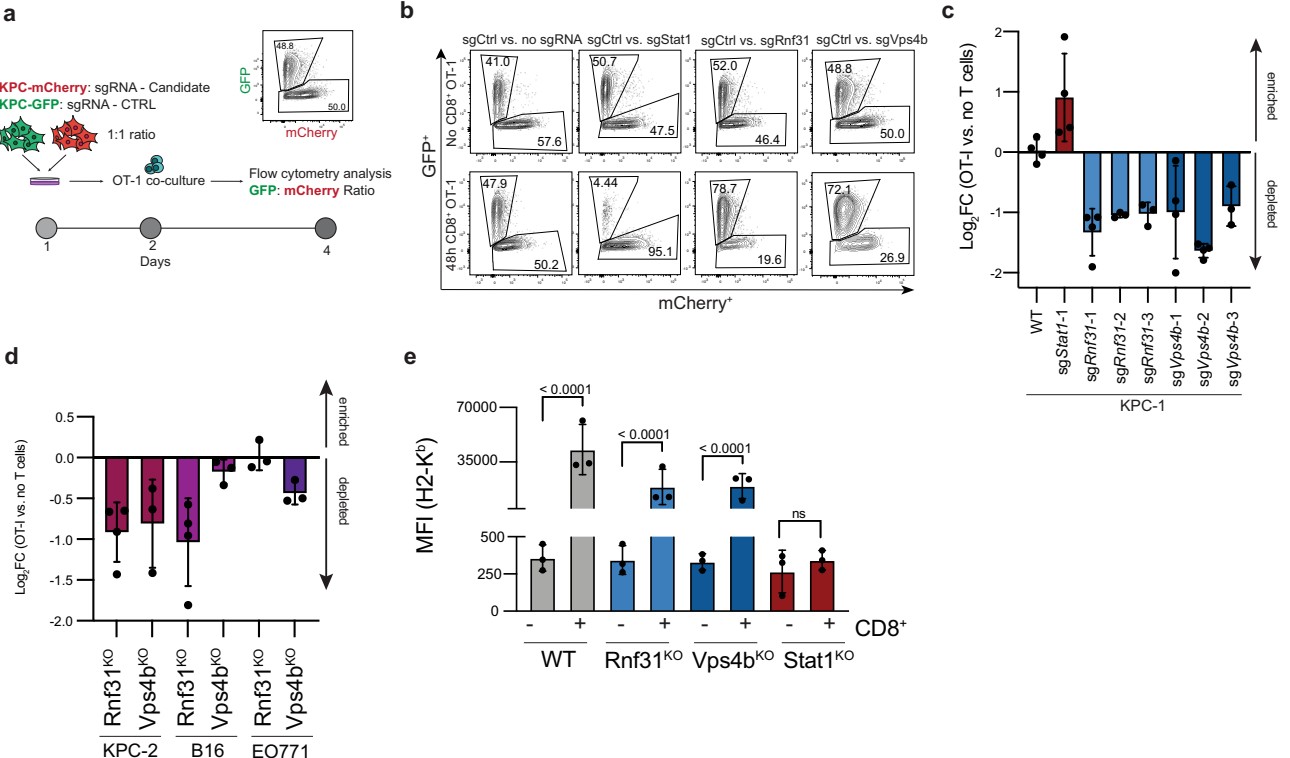

**Fig. 3 Arrayed validation of selected screening hits in vitro. a** Schematic of in vitro competition assay. Tumor cell:T cell co-cultures were carried out with an E:T ratio of 1:1 in all experiments. **b** Representative flow cytometry plots (GFP vs. mCherry) of the arrayed hit validation with and without T cell co-culture. **c** Log$_2$ Fold change of mCherry$^+$ KPC population before and after OT-I co-culture with different sgRNAs targeting candidates *Rnf31* and *Vps4b*. $n = 3$ for Rnf31-2; Rnf31-3 and Vps4b-3, all other conditions $n = 4$. **d** Log$_2$ Fold change of mCherry$^+$ KPC population before and after OT-I co-culture in an alternative KPC cell line, B16-F10 melanoma cells and EO771 breast cancer cells. $n = 4$ for KPC-2 Rnf31 and B16-F10 Rnf31, all other conditions $n = 3$. **e** Mean fluorescence intensity (MFI) of Pan-H2-K$^b$. $n = 3$ independent experiments. Significance was determined with a one-way ANOVA analysis. Non-significant, $p > 0.05$. Values represent mean±SD, data are derived from three independent experiments. Source data are provided as a Source Data file.

cells commonly sequestrate MHC-I to evade the adaptive immune system[9], prompting us to suspect that loss of *Rnf31* and *Vps4b* facilitates CD8$^+$ mediated killing by increasing MHC-I levels on PDA. To test this hypothesis, we assessed surface MHC-I levels in KPC cells upon exposure to CTLs. Confirming our assay, we observed a robust induction of MHC-I upregulation in parental KPC cells, which, as expected, was perturbed in IFNγ signaling deficient *Stat1*$^{KO}$ cells (Fig. 3e, Supplementary Fig. 3h). Next, we analyzed MHC-I induction in *Rnf31*$^{KO}$ and *Vps4b*$^{KO}$ KPC cells. However, we observed similar surface MHC-I levels compared to the parental cell line (Fig. 3e), demonstrating that enhanced CTL-mediated killing is not triggered by an increase in antigen presentation.

Next, we sought to gain insights into the transcriptional networks mediating the sensitizing effects of *Rnf31*$^{KO}$ and *Vps4b*$^{KO}$ to CTL killing. We, therefore, performed RNA-sequencing (RNA-seq) on the different PDA knock-out lines with and without six hours of CD8$^+$ T cell exposure (Supplementary Fig. 3b). Confirming functional gene knock-outs, transcript levels of *Rnf31* and *Vps4b* were downregulated in the respective PDA lines (Supplementary Fig. 3d). Furthermore, *Rnf31*$^{KO}$ and *Vps4b*$^{KO}$ cell lines displayed relatively mild, but consistent transcriptional changes (Supplementary Fig. 3f). Among the differentially expressed genes in CTL-treated *Rnf31* or *Vps4b* knock-out PDA cells were several cytokines and chemokines, including the Cxcr3 ligands *Cxcl9/10/11* (Supplementary Fig. 3f, g). Notably, the Cxcr3-Stat3 signaling axis has previously been described to enhance PDA aggressiveness and to contribute to an immune suppressive environment through

inducing PD-L1 (CD274) expression[25,26], and low expression of these chemokines is correlated with a better prognosis in human PDA patients (Supplementary Fig. 3i). However, differences in cytokine expression alone could not explain the in vitro phenotypes observed in the KPC-CTL co-culture screen, prompting us to conduct further mechanistic studies on *Rnf31* and *Vps4b* in KPC cells.

**Rnf31 loss sensitizes PDA to TNF-induced apoptosis via caspase 8.** Cytotoxic T cells induce death in target cells via different processes, including the release of TNF, secretion of granules filled with granzymes and perforins, and by engaging the Fas-FasL axis. To systematically explore which of these effector mechanisms are sensitized upon *Rnf31* and *Vps4b* inhibition, we first assessed tumor cell sensitivity to TNF ligands. Interestingly, we found that parental KPC cells and *Vps4b*$^{KO}$ KPC cells were insensitive to TNF-induced apoptosis, but that *Rnf31*$^{KO}$ KPC cells rapidly underwent cell death upon TNF treatment (Fig. 4a). Engagement of the TNF receptor triggers several signaling branches, including pro-survival NFκB signaling as well as apoptosis induction via caspase 8 cleavage[27–29]. Notably, *Rnf31* has previously been reported to function as an E3 ubiquitin-protein ligase within the LUBAC, which is involved in regulating NFκB signaling[27]. We, therefore, speculated that the *Rnf31* knock-out sensitizes tumor cells to TNF-mediated apoptosis either indirectly, by abrogating NFκB pro-survival signaling, or directly, by facilitating caspase 8 cleavage. When we first assessed TNF-mediated NFκB activation, we found phosphorylation of the

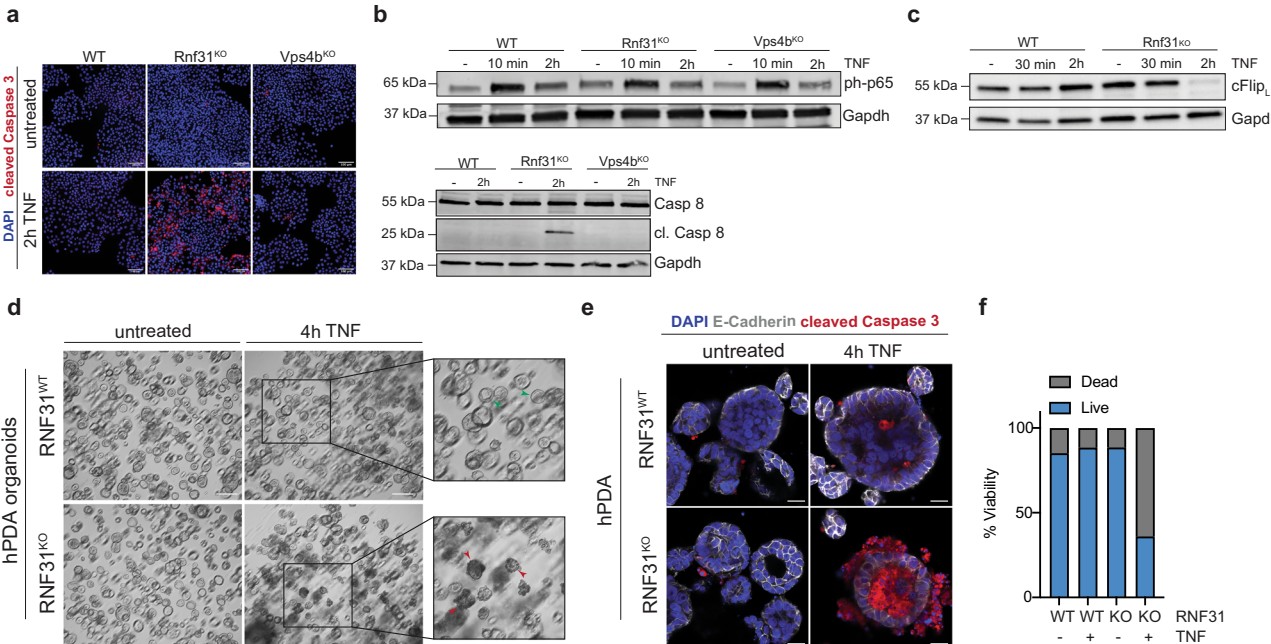

**Fig. 4 *Rnf31*[KO] sensitizes PDA to TNF-triggered apoptosis. a** Immunofluorescence staining of KPC-1 candidate lines after 2 h of 100 ng/ml TNF. Cleaved caspase 3 (red) and DAPI (blue). Scale bar represents 100 μm. **b** Western Blot analysis of KPC cell lines after TNF treatment (100 ng/ml) for active NFκB signaling (phospho-p65; upper panel) and cleaved caspase 8/caspase 8 (bottom panel). Gapdh was included as loading control. **c** Western Blot analysis of KPC WT and Rnf31[KO] cell lines after TNF treatment (100 ng/ml) overtime for c-Flip$_L$. Gapdh was included as loading control. Western plots in (**b**, **c**) were repeated three times independently. Representative images are shown. **d** Brightfield images of human PDA organoids in the presence of 100 ng/ml TNF for 4 h. Boxes highlight viable (green arrows) and dying organoids (red arrows). Scale bar represents 200 μm. **e** Whole-mount staining of human PDA organoids after 4 h TNF (100 ng/ml) treatment with cleaved caspase 3 (red), E-cadherin (white), and DAPI (blue). Scale bar represents 20 μm. Immunofluorescence stainings and bright-field images in (**a**), (**d**), and (**e**) were repeated twice on independent samples. Representative images are shown. **f** Relative organoid viability, individual organoids were counted and classified in 'live' or 'dead'. Percentage dead/live of the counted area of all organoids is displayed. Values represent mean, *n* = 2 independent experiments. Source data are provided as a Source Data file.

NFκB subunit p65/Rela in all genetic backgrounds, including *Rnf31*[KO] cells (Fig. 4b), indicating functional NFκB signaling. When we next analyzed caspase 8 cleavage upon TNF treatment, activation was observed in *Rnf31*[KO] KPC cells but not in parental KPC- or *Vps4b*[KO] KPC cells (Fig. 4b). Hence, our data suggest that in Rnf31-deficient cells intact NFκB signaling is not sufficient to rescue TNF-activated caspase 8 cleavage.

A recent study proposed that Rnf31 is involved in stabilizing the anti-apoptotic protein c-Flip[27], prompting us to assess whether loss of *Rnf31* sensitized KPC cells to caspase 8-mediated apoptosis via destabilization of c-Flip. In line with this hypothesis, we found that c-Flip was degraded in *Rnf31*[KO] KPC cells but not in parental KPC cells upon TNF exposure (Fig. 4c). Moreover, *Cflar*, the gene encoding for c-Flip, was among the most strongly depleted genes in our genome-wide screen (Fig. 1d), and *Cflar*[KO] KPC cells mimicked the *Rnf31*[KO] phenotype and underwent apoptosis after TNF exposure (Supplementary Fig. 4d).

To next assess whether loss of Rnf31 also sensitizes human PDA to TNF-mediated cell death, we generated patient-derived and engineered human pancreatic cancer organoids (hPDA) with *RNF31*[KO] mutations and treated these organoids for four hours with TNF. In line with results from murine PDA tissues, only *RNF31*[KO] but not *RNF31*[WT] PDA organoids activated apoptotic cell death upon TNF stimulation (Fig. 4d–f, Supplementary Fig. 4a–c). Taken together, our results suggest that loss of the LUBAC subunit Rnf31 sensitizes murine and human pancreatic cancer to CTL killing by rendering cells susceptible to caspase-8-mediated apoptosis upon TNF signaling (Supplementary Fig. 4e).

**Vps4b depletion impairs functional autophagy and increases intracellular Granzyme B levels.** Vps4b is part of the ESCRT-III complex functions as an AAA-type ATPase involved in diverse processes, including the catalyzation of phagophore closure during autophagy[30]. Considering that several autophagy-related genes have been identified as sensitizers for CTL-mediated killing, we reasoned that *Vps4b*[KO] cells might sensitize PDA to CTL-mediated killing by inhibiting autophagy. To test this hypothesis, we transduced cells with an autophagic flux reporter, and assessed if autophagy is impaired in *Vps4b* knock-out KPC cells[31]. The reporter consists of a LC3-GFP-LC3ΔG-RFP fusion protein; LC3-GFP is localized to the autophagosome and degraded during autophagy, and LC3ΔG-RFP lacks a C-terminal glycine and stably resides in the cytoplasm during autophagy (Fig. 5a). While parental KPC cells showed a strong upregulation of autophagy upon starvation (Fig. 5b), *Vps4b*[KO] KPC cells showed an impaired autophagic flux, similar to fully autophagy-deficient *Atg5*[KO] KPC cells (Fig. 5b). These data suggest that loss of Vps4b sensitizes PDA to CTL killing through disrupting autophagy. To test whether this mechanism is also conserved in human cells, we next measured the autophagic flux in HEK293T cells upon *VPS4B* deletion. However, in contrast to the loss of function mutation in *ATG9A*, *VPS4B* depletion did not impair autophagy induction (Supplementary Fig. 5d), and we speculate that in human cells homologs of *VPS4B* might lead to functional compensation upon gene disruption.

In previous studies, several different mechanisms have been proposed of how autophagy could modulate immune evasion. One study suggested that high autophagy rates in PDA may

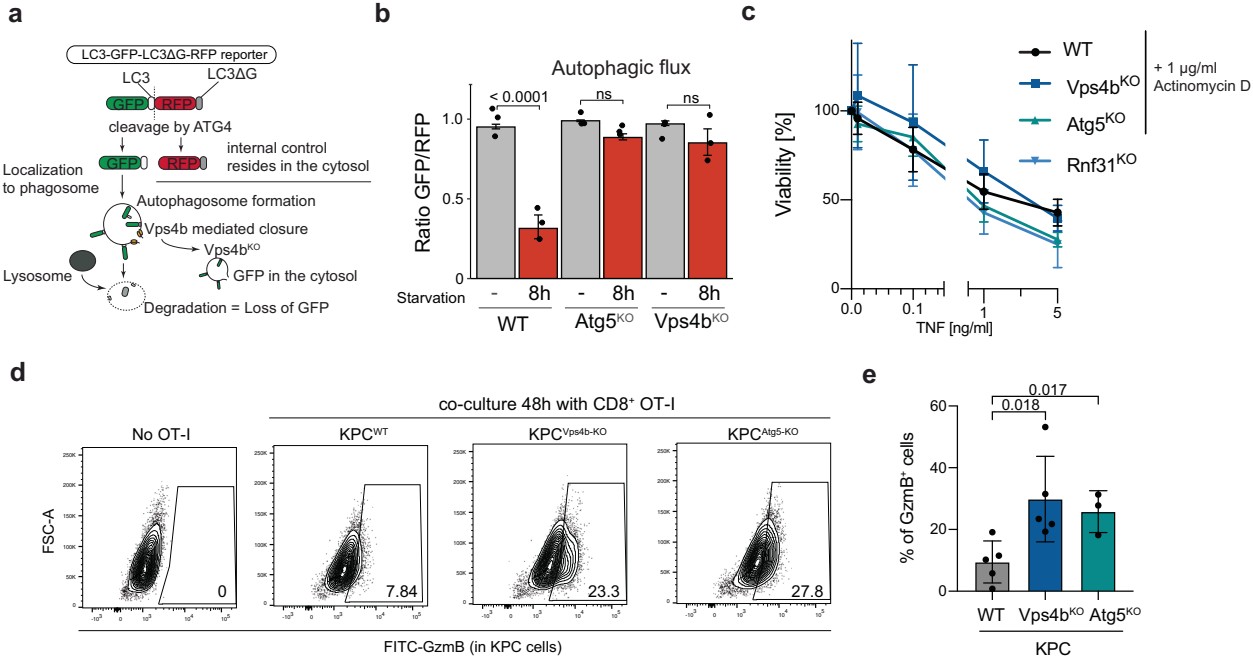

**Fig. 5 Disruption of Vps4b leads to impaired autophagy and granzyme B accumulation in tumor cells. a** Schematic of autophagic flux reporter based on the LC3-GFP-LC3ΔG-RFP probe adapted from Kaizuka et al.[31]. **b** Quantification of autophagic flux by flow cytometry in different KPC lines under normal and starvation conditions (8 h in PBS + 2% FBS). Bars represent the ratio of GFP to RFP expressing cells with $n = 3$ independent experiments. **c** Assessment of TNF sensitivity threshold in different KPC cells in the presence of 1 μg/ml actinomycin D, except for $Rnf31^{KO}$; here no Actinomycin D was present. Crystal violet staining was used for the quantification of viable cells. Represented data are relative to untreated control cells. Data are from $n = 4$ independent experiments. **d** Flow cytometric analysis of intracellular granzyme B in KPC-1 cells. Cells were gated on FSC/SCC—viability—CD8- negative. **e** Quantification of granzyme B positive cancer cells and T cells based on (**d**) $n = 5$ for WT and Vps4b and $n = 3$ for Atg5. Significance in (**b**) was determined with one-way ANOVA. Significance in (**e**) was determined with an unpaired, two-tailed $t$-test. ns, non-significant, $p > 0.05$. Values represent mean ± SD, data are derived from at least three independent experiments. Source data are provided as a Source Data file.

contribute to immune evasion by sequestering surface MHC-I levels[9]. In KPC cells, nevertheless, we observed a robust induction of MHC-I surface expression upon CTL exposure, which was also not affected by Vps4b depletion (Fig. 3e). Another study suggested that autophagy inhibition may facilitate CTL-mediated killing by increasing sensitivity to TNF-induced cell death[11]. However, in KPC cells mutant for *Vps4b* or *Atg5* we did not observe apoptosis induction upon TNF treatment (Fig. 4a, b), and pre-treatment of cells with Actinomycin D, a NFκB-mediated sensitizer to TNF-induced apoptosis, revealed similar sensitivity of *Vps4b*^KO-, *Atg5*^KO- and parental KPC cells to increasing concentrations of TNF (Fig. 5c, Supplementary Fig. 5b). A third study found that high autophagy levels in breast cancer cells promoted NK cell-derived granzyme B degradation, suggesting that this mechanism could contribute to the resistance to CTL killing[32,33]. We, therefore, hypothesized that autophagy deficiency could sensitize PDA cells to CTL killing through insufficient granzyme B clearance, and quantified intracellular granzyme B levels in KPC cells upon OT-I T cell exposure. Indeed, while OT-I T cells produced comparable amounts of granzyme B, autophagy-deficient *Atg5*^KO- and *Vps4b*^KO KPC cells accumulated more granzyme B than parental KPC cells (Fig. 5d, e, Supplementary Fig. 5c). Taken together, our data suggest that *Vps4b* inhibition perturbs autophagy in KPC cells, which in turn reduces their capability to degrade granzyme B upon CTL-mediated killing.

**Rnf31 and Vps4b inhibition increases CTL infiltration and effector function in vivo.** To further characterize increased CTL susceptibility of *Rnf31*^KO and *Vps4b*^KO KPC cells in vivo, we analyzed the effect of these mutations on PDA progression and the tumor microenvironment in mice. Therefore, we orthotopically transplanted KPC-Cas9 cells with different genotypes (wildtype, *Rnf31*^KO and *Vps4b*^KO) into immune-competent C57BL/6 animals, and assessed survival and tumor weight, as well as immune cell composition and effector function using flow cytometry (Fig. 6a, Supplementary Fig. 6d). While remaining Cas9 expression in these lines did not affect tumor growth (Supplementary Fig. 6a), loss of *Rnf31* and *Vps4b* markedly decreased tumor mass and resulted in significantly enhanced survival of tumor-bearing mice (Fig. 6b). In case of *Vps4b*^KO tumors the effect was strongly dependent on adaptive immunity, since tumor mass reduction was not apparent in RAG1$^{-/-}$ mice (Supplementary Fig. 6b). *Rnf31*^KO tumors, however, also showed reduced growth compared to KPC^WT tumors in RAG1-deficient hosts, most likely due to the continued expression of TNF and other death receptor ligands by NK cells (Supplementary Fig. 6b). We next analyzed immune cell infiltration and CD8$^+$ T cell effector function across the different tumor genotypes. While the loss of *Vps4b* and *Rnf31* did not cause significant changes in macrophages, CD11c$^+$ dendritic cells, CD4$^+$ T helper cells, NK cells, and CD4$^+$ Foxp3$^+$ regulatory T cells, we detected a minor increase of neutrophils in *Rnf31*^KO tumors (Supplementary Fig. 6c). In addition, we observed a substantial increase of infiltrating CD8$^+$ T cells in *Vps4b*^KO and *Rnf31*^KO tumors compared to parental KPC tumors (Fig. 6c). Moreover, in *Vps4b*^KO and *Rnf31*^KO tumors infiltrating CD8$^+$ T cells displayed a reduction in PD1 surface levels and intracellular Eomes levels, as well as increased effector cytokine production (i.e., TNF and IFNγ), indicating a reduction in T cell exhaustion (Fig. 6c). Notably, we also observed a decrease in actively proliferating CD8$^+$ Ki67$^+$ tumor-infiltrating T cells in *Rnf31*^KO and *Vps4b*^KO tumors. While

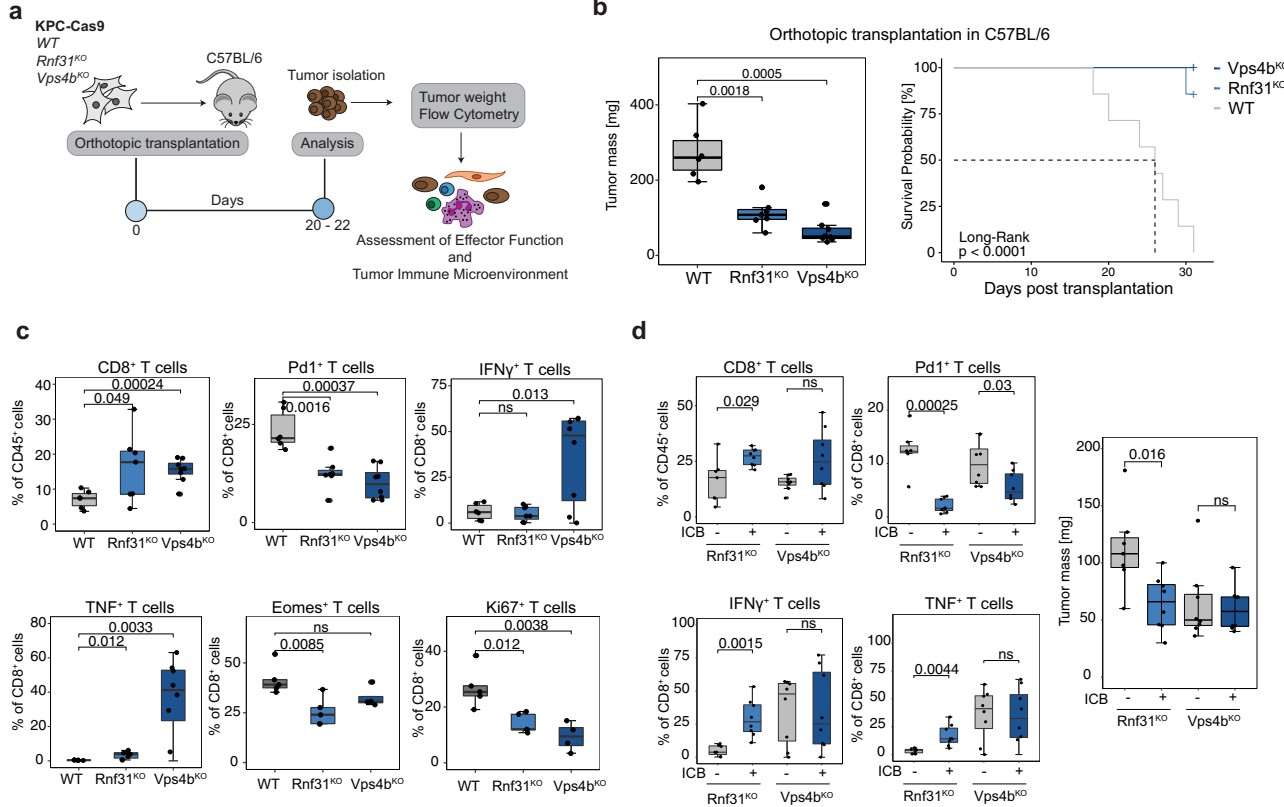

**Fig. 6 *Rnf31*^KO and *Vps4b*^KO enhances CD8^+ T cell function in vivo. a** Schematic of in vivo experimental set up. **b** Tumor weight (left panel; *n* for WT = 6; *Rnf31*^KO = 7; *Vps4b*^KO = 8) and survival (right panel; *n* = 7 per group) after orthotopic transplantation into C57BL/6 mice. Log-Rank test was performed to assess statistical significance in Kaplan–Meier plot. **c** Flow cytometry analysis of effector function of CD8^+ T cells within tumors; *n* in WT = 6; *Rnf31*^KO = 7; *Vps4b*^KO = 8; panels Eomes^+ and Ki67^+ *n* = 5 per group for all genotypes). **d** Assessment CD8^+ T cell effector function and tumor weight (right panel) in Rnf31^KO or Vps4b^KO tumors with or without immune checkpoint inhibition (ICB); *n* = 8 per group). Significance in (**b**, left panel), (**c, d**) was determined with unpaired two-tailed *t* tests; ns, non-significant, *p* > 0.05. The middle line in the boxplots shows the median, the lower and upper hinges represent the first and third quartiles, and whiskers represent ±1.5× the interquartile range. Source data are provided as a Source Data file.

at first sight, this finding seems counterintuitive, it could be explained by the decreased tumor load compared to the parental KPC tumors at the stage of analysis (see reduced tumor mass in Fig. 6b). Finally, we also performed immune checkpoint blockade (ICB) treatment in mice transplanted with *Rnf31*^KO and *Vps4b*^KO KPC tumors, and found that CD8^+ T cell infiltration was further increased and accompanied by markedly reduced PD1 surface expression (Fig. 6d). In *Rnf31*^KO tumors ICB also potentiated effector cytokine production and led to significant tumor mass reduction (Fig. 6d). Together, our findings from in vivo experiments demonstrate that loss of *Rnf31* and *Vps4b* sensitize PDA to CTL-mediated killing not only in a cell-autonomous manner, but also in a non-cell-autonomous manner through increasing CTL effector function.

## Discussion

Several recent studies performed CRISPR screening in PDA to study metastasis formation, metabolic vulnerabilities, combinatorial drug targeting, and therapy resistance[21,34–36]. Here, we applied in vitro and in vivo CRISPR screening in PDA to interrogate tumor intrinsic mechanisms of immune evasion. One of the strongest sensitizers to CTL-mediated killing was *Rnf31*, for which we show that its inactivation facilitates TNF-induced apoptosis via caspase 8 cleavage in murine and human PDA. Notably, previous work has already linked TNF resistance to immune evasion[11,37]. However, these studies have been conducted in a TNF-susceptible colorectal cancer cell line (MC38,

impeding the identification of TNF sensitizers such as *Rnf31*. In contrast, KPC pancreatic cancer cells are intrinsically resistant to TNF, which allowed us to unravel a mechanism that abates TNF resistance. We further show that TNF is sufficient to induce apoptosis in Rnf31 deficient pancreatic cancer cells, in contrast to B16 melanoma cells where IFNγ is required for TNF-induced apoptosis after Rnf31 disruption[38].

Another strong sensitizer to CTL-mediated killing identified in our screen was *Vps4b*, which could be linked to autophagy. Autophagy has recently been postulated as an important modulator of anti-tumor immunity in several cancer entities[9–11,39]. However, in contrast to previous findings, we did not observe increased MHC-I antigen presentation[9] or enhanced TNF-induced apoptosis[10,11] upon autophagy inhibition. Instead, we provide evidence that impaired autophagy leads to reduced granzyme B clearance, suggesting that *Vps4b* or *Atg5* depletion facilitates tumor cell lysis by CD8^+ T cells through enhanced granzyme B accumulation. While *VPS4B* depletion did not perturb autophagy in human cells, in the future it would be interesting to test whether autophagy inhibition also leads to elevated granzyme B accumulation in human cancers. Moreover, since the administration of autophagy inhibitors alone or in combination with conventional chemotherapy did not improve progression-free- or overall survival in previous clinical studies[40], it will be interesting to see if more recent trials in which autophagy inhibition is combined with immunotherapy will report higher efficacy (LIMIT; NCT04464759).

We identified *Rnf31* and *Vps4b* in our in vitro and in vivo CRISPR screens using the OVA/OT-I tumor model, which

ensured sufficient antigenicity of tumor cells. Importantly, arrayed in vivo validation using orthotopic transplantation of Rnf31KO- and Vps4bKO KPC tumors into immune-competent C57BL/6 showed that the observed phenotype was conserved without employing the OVA/OT-I system, strengthen potential translatability of our findings to the clinics. However, as T cell immunity is generally dependent on sufficient tumor antigenicity, future studies should determine whether Rnf31 and Vps4b depletion will show similar phenotypes in other PDA tumor models. Nevertheless, Rnf31 is a promising candidate in this context, as it acts through TNF-mediated apoptosis that is independent of antigen presentation. One limitation of our study is that our experimental set up relied on orthotopic transplantation, which does not faithfully recapitulate PDA stroma formation[41]. Therefore, we exclusively focused on tumor cell–immune cell interactions and did not analyze potential changes in the tumor stroma in this study. As the tumor stroma is known to be an important determinant for PDA prognosis and treatment[42,43], addressing the effect of Rnf31 and Vps4b depletion in the context of the tumor stroma would be highly valuable.

Taken together, we used functional genomic approaches to identify mechanisms for circumventing immune evasion in PDA. Analysis of two of the strongest hits, Vps4b, and Rnf31, demonstrated that their inhibition sensitizes tumor cell clearance directly, via cell-autonomous mechanisms, and indirectly, by increasing the number and functionality of intratumoral CD8+ T cells. Our insights in sensitizing pancreatic cancer to the host immune system could open up promising strategies to enhance the efficacy of T cell-mediated tumor killing, potentially allowing PDA patients to benefit from the vast advances made in the field of cancer immunotherapy in the future.

## Methods

**Animals**. Wildtype C57BL/6 mice were obtained from Charles River Laboratories. RAG1−/− (NOD.129S7(B6)-Rag1tm1Mom/J) and OT-I (C57BL/6-Tg(TcraTcrb) 1100Mjb/J) were obtained from Jackson Laboratories and bred in-house. For all experiments, male mice between 8 and 12 weeks were used. All animals were housed in a pathogen-free animal facility in cages with up to five animals at the Institute of Molecular Health Sciences at ETH Zurich and kept in a temperature- and humidity-controlled room on a 12 h light–dark cycle. All animal experiments were performed in accordance with protocols approved by the Kantonales Veterinäramt Zurich in compliance with all relevant ethical regulations.

**Cell culture**. The KPC-1 cell line (C57BL/6 background) was generated by Dr. Jen Morton (Beatson Institute) and purchased at Ximbio (Cat# 153474). KPC cells were derived from primary KPC tumors obtained from Pdx-Cre; KrasG12D/+; Trp53R172H/+ mice. The KPC-2 cell line (C57BL/6 background) was generated in-house from p48-Cre; KrasG12D/+; Trp53fl/fl mice. The melanoma cell line B16, the breast cancer cell line EO771, and HEK293T cells were obtained from ATCC. All cell lines used in this study were cultured in Iscove's Modified Dulbecco's Medium (IMDM, 31980030, Gibco) supplemented with 10% fetal bovine serum (FBS), 1% penicillin/streptomycin (Gibco) and 50 μM β-Mercaptoethanol (Gibco). Cells were incubated at 37 °C in 5% $CO_2$. The parental cancer cell lines were engineered with Lenti-Cas9-Hygromycin and Lenti-Ovalbumin-mCherry-Blasticidin constructs (for details see "Plasmids") in order to express Cas9 and full-length Ovalbumin (KPC-Cas9-OVA). KPC cell line (KPC-1) was received from Ximbio (Cat# 153474), including a materials transfer agreement.

**Plasmids**. For generation of Lenti-EF1α-Cas9-P2A-HygromycinR we replaced Blasticidin from the original vector (Addgene #52962) by HygromycinR using Gibson Assembly. For generation of Lenti-EF1α-Ovalbumin-P2A-mCherry-P2A-BlasticidinR, we replaced Cas9 (Addgene #52962) by Ovalbumin-P2A-mCherry derived from the original vector (Addgene #113030) with Gibson Assembly. The LC3-GFP-LC3ΔG-RFP construct was obtained from Addgene (#84572) and cloned into Lenti-EF1α-Cas9-P2A-BlasticidinR by replacing Cas9-P2A-BlasticidinR. The following vectors LentiGuide-Puro (#52963), LentiCRISPRv2-puro (#98290) and pLenti-PGK-Hygro-KRAS(G12V) (#35635) were obtained from Addgene.

**Lentivirus production**. For lentivirus production of CRISPR libraries, transgenes or single guide HEK293T cells were transfected with, HEK293T cells (ATCC) were seeded in T175 cell culture flasks in DMEM (Gibco) supplemented with 10% FBS and 1% Penicillin/Streptomycin and grown up to 70% confluency. HEK293T cells

were transfected with the following mixture: 10.4 μg psPAX2-Plasmid, 3.5 μg pMD2.G, 13.8 μg lentiviral vector of interest (see section "Plasmids") in a volume of 1000 μl Opti-MEM (Gibco) in tube 1. In a second tube, 138 μl 1 mg/mL PEI (Polysciences) was mixed with 862 μl Opti-MEM. Both tubes were incubated at room temperature for 5 min, mixed, and incubated again 20 min at room temperature and added to the cells in the evening. The next morning, the medium was refreshed. After 48 and 72 h, the supernatant was harvested, filtered with 0.45 μm syringe filters (Sarstedt) and concentrated by centrifugation at $24,000 \times g$ for 2 h. Plasmids psPAX2 (Addgene #12260) and pMD2.G (Addgene #12559) were gifts from Didier Trono.

**Genome-wide CRISPR screen**. The genome-wide Brie CRISPR-KO library (4 sgRNAs per gene; ~80,000 sgRNAs) was purchased from Addgene (#73632) and amplified according to the supplier's protocol. KPC-Cas9-OVA were infected with the lentiviral Brie CRISPR-KO library at a multiplicity of infection (MOI) of 0.3 while keeping a 500x coverage of the library. One day post transduction, cells were selected with 2 μg/ml Puromycin for 5 days in order to select for successfully transduced cells and to allow gene CRISPR-mediated gene knockout. For OT-I T-cell co-culture, $4 \times 10^7$ KPC cells were plated at a final confluency around 60% and incubated for 3 days together with activated T-cells at an E:T ratio of 1:1. After T-cell killing, surviving KPC cells were left to recover for another 3 days. Untreated KPC-Cas9-OVA-Brie cells were cultured alongside and harvested for DNA isolation together with OT-I treated cells. For each replicate and sample, DNA was isolated from $4 \times 10^7$ KPC cells using the Blood & Cell Culture DNA Maxi Kit (Qiagen). NGS libraries were prepared using the following primers:

Staggered P5 forward primer:
5′AATGATACGGCGACCACCGAGATCTACACTCTTTCCCTACACGACGC TCTTCCGATCT[N1-8]T TGTGGAAAGGACGAAACACCG

Barcoded P7 reverse primer:
5′CAAGCAGAAGACGGCATACGAGATNNNNNNNNNGTGACTGGAGTTCA-GACGTGTGCTCTT CCGATCTTCTACTATTCTTTCCCTGCACTGT

For each sample, total DNA obtained from $4 \times 10^7$ cells was used as input with 10 μg DNA per 100 μl PCR reaction. DNA was amplified using Herculase II Fusion DNA Polymerase (Agilent) according to the manufacturer's conditions with 2 μl Herculase II and 2.5 mM $MgCl_2$. Annealing was performed at 55 °C and a total of 24 cycles. PCR reactions were cleaned up using 0.8× AMPure beads (Beckman Coulter). NGS libraries were run on the Illumina NovaSeq 6000 System generating 100 bp single-end reads.

**Data analysis**. Demultiplexed reads were trimmed to exact 20 bp (sgRNA) using cutadapt. Subsequently, read counts were assessed using MAGeCK (v0.5.6) as well as sgRNA enrichment/depletion with "mageck test -k *readcouts.txt* -t OT-I -c CTRL --norm-method control --control-sgrna *nontargeting*.txt". For MAGeCK analysis three biological screening replicates were pooled. For pathway analysis of gene enrichment/depletion the cut-off was set at FDR < 0.1 and GO term analysis for candidates was performed using the Molecular Signature Database (MSigDB). The screening data set can be found in Supplementary Tables 1 and 2 and via the GEO accession number GSE180834.

**Isolation and activation of CD8 T-cells**. CD8+ OT-I T-cells were isolated from spleen, axillary, and inguinal lymph nodes from OT-I mice. CD8+ cells were enriched using magnetic beads for MACS (130-104-075, Milteny Biotec). T-cells were cultured in IMDM supplemented with 10% FBS, 1% penicillin/streptomycin (Gibco), 50 μM β-Mercaptoethanol (Gibco), and 100 ng/ml Il-2 (Peprotech). Cells were kept at 37 °C in 5% $CO_2$. After T-cell isolation, cells were activated for 24 h using 2 μg/ml of anti-Cd28 and anti-Cd3ε antibodies (102116 and 100340, Bio-Legend; 1:500 each).

**In vitro cytotoxicity assays and CRISPR-KO generation**. One day prior to OT-I co-culture, a total of $5 \times 10^4$ Ovalbumin-expressing KPC cells were plated into 24-well plates. For competition assays, KPC-Cas9-OVA-mCherry (with CRISPR-KO of the indicated gene) and KPC-Cas9-OVA-EGFP-sgRNA@ctrl were plated in a 1:1 ratio. Activated OT-I T-cells were added in the presence of 100 ng/ml Il-2 for one to two days, depending on downstream analysis. EGFP:mCherry ratio was assessed using flow cytometry. In brief, cells were trypsinized, spun down, and washed in FACS buffer (2% FBS, 2 mM EDTA in PBS). For MHC class I assessment, cells were detached using 5 mM EDTA. The following antibodies were used: CD8a-APC (1:600; 53-6.7; eBioscience), CD8a-FITC (1:600; 53-6.7; eBioscience), H-2Kb-APC (1:100; AF6-88.5.5.3; eBiosciences). SYTOX Blue was used as a viability dye. For intracellular Granzyme-B staining (GzmB-FITC; GB11; BioLegend, 1:200) cells were stained with eFluor780 (fixable viability dye, 1:2000) and stained with 4% formalin before permeabilization. GzmB-FITC staining was carried out in permeabilzation buffer for 30 min at room temperature. The following sgRNAs were cloned into the LentiGuide-puro vector and stably integrated into target cells:

Ctrl-1: 5′ GCGAGGTATTCGGCTCCGCG
Rnf31-1: 5′ CTACCTCAACACCCTATCCA
Rnf31-2: 5′GATGGATTGAGTTTCCCCGA
Rnf31-3: 5′ GAACTATGAGTTGTTGGACG
Vps4b-1: 5′ TAAAGCCAAGCAAAGTATCA

Vps4b-2: 5′ GGCTGCACGGAGAATTAAGA
Vps4b-3: 5′ GGAAAGCGGACACCTTGGAG
Cflar-1: 5′ TGGGTTATGTCATGTGACTT
Vps4a-1: 5′ ACTCACATTTGATGGCGTGG
Vps4a-2: 5′ CTTAGACATCAGATCCGAGG
Vta1-1: 5′ TCATAACTGCTTAAAGAATG
Vta1-2: 5′ CACAGTAGTAGGCCACCACG
Chmp4b-1: 5′ GGGCGGCCCGACCCCCCAGG
Chmp4b-2: 5′ CAGCACATACATGTTGTCGT
Stat1: 5′ GGATAGACGCCCAGCCACTG
Atg5: 5′ AAGAGTCAGCTATTTGACGT

For ESCRT-III sgRNAs both sgRNAs were transduced in a pool. For RNA-Seq and in vivo studies, KPC-1-Cas9-Hygromycin cells were transduced with a pool of sgRNA1-3 targeting Rnf31 or Vps4b. For alternative cell line validation KPC-2, B16, and EO771 cells were transduced with a pool of sgRNA1-3 targeting Rnf31 or Vps4b. Cells were selected with 2 μg/ml Puromycin for 3–5 days.

**Autophagy flux assay.** LC3-GFP-LC3ΔG-RFP construct was stably integrated into KPC-Cas9-OVA cells or HEK293T cells and GFP+/RFP+ single-cell clones were sorted and expanded. Cells were starved for 8 h or overnight (HEK293T cells) in 2% FBS/PBS or 2% FBS/EBSS (HEK293T cells) at 37 °C. Subsequently, cells were collected for flow cytometry analysis in order to assess GFP and RFP expression. The autophagic flux was assessed by calculating the GFP/RFP ratio and comparison to the non-starved control condition. HEK293T cells were engineered for ATG9A-KO or VPS4B-KO with the following sgRNAs, cloned into LentiCRISPRv2-Puromycin, and lentivirally transduced in order to generate knockouts.

ATG9A: 5′ CCTGTTGGTGCACGTCGCCG
VPS4B-1: 5′ ATGTCACCTGTAAAAAGATG
VPS4B-2: 5′ CAGCGCAAGAAGACAAGGCT

sgRNAs targeting VPS4B were transduced together in a lentiviral pool.

**Human pancreatic cancer organoids.** Human pancreatic tissue was obtained from pancreatic islet isolation procedures from the University Hospital Zürich; human PDA tissue was obtained from surgical tumor resections at the University Hospital Zurich. The Clinical Ethics Committee of the University Hospital Zurich approved the use of the samples for the generation of tumor organoids lines, and informed consent was provided from all patients. For Normal human pancreatic organoids (hPan) and pancreatic cancer organoids (hPDA) were generated as described elsewhere[44]. Expansion medium (EM) contained Advanced DMEM/F12 supplemented with 10 mM HEPES, 1× Glutamax, 1% Penicillin/Streptomycin, 1x B27 without vitamin A (all Gibco), 1.25 mM N-acetylcysteine (Sigma), 25% WNT3A-conditioned medium (CM), 10% RSPO1-CM, 10% NOGGIN-CM (all CM produced in-house), 10 mM Nicotinamide (Sigma), 50 ng/mL human EGF (Peprotech), 100 ng/ml FGF10 (Peprotech), 10 nM Gastrin (Tocris Bioscience), 0.5 μM TGF-b type I receptor inhibitor A83-01 (Tocris Bioscience) and 1 μM PGE2 (Tocris Bioscience). Organoids were split every 7–10 days using TrypLE Express (Gibco) and fire-polished Pasteur pipettes in a 1:3–1:4 ratio. After passaging, organoids were plated in 20 μl drops of Matrigel (Corning) and overlaid with EM supplemented having 10 μM RhoKinase inhibitor (Y-27632; Abmole). Organoids from healthy donors were lentivirally transduced to express oncogenic KRAS^G12V and to knockout TP53 (hPan-KP). Both, hPan-KP and hPDA organoids were transduced with LentiCRISPRv2-puro to knockout RNF31. For lentiviral transduction, 3–4 full drops of organoids per condition were processed into single cells, mixed with 500 μl EM + 10 μM RhoKinase inhibitor + 50 μl concentrated virus and spun for one hour at 32 °C at 300 × g in a 24-well plate. After 3–4 h of incubation at 37 °C, cells were collected and plated in Matrigel. Organoid were selected with 1.5 μg/ml puromycin (RNF31-KO), 10 μM Nutlin-3a (TP53-KO; Sigma) and 300 μg/ml hygromycin (KrasG12V). The following sgRNAs were used— TP53: 5′ GAAGGGACAGAAGATGACAG, RNF31-1: 5′ CCACCGTGCTGCG AAAGACA, RNF31-2: 5′ CCCAACCCCTTACAGCCTCG, RNF31-3: 5′ GGA TCATGCTCACTAGCTGG.

**Sublibrary generation.** For sublibrary generation, the top candidates (enriched and depleted; FDR < 10%) were selected (Supplementary Table 3). If there were many genes within one pathway, only a couple of genes were selected to avoid redundancy. For each gene of the gene list (63 candidates and Ovalbumin) 10 sgRNAs (7 sgRNAs for Ovalbumin) were designed with the GPP sgRNA designer (Broad Institute). A total of 600 non-targeting controls were likewise included. Oligonucleotides having BsmBI (EspI) restriction sites, a single guide RNA sequence as well as primer binding sites for oligo pool amplification were synthesized (Twist Bioscience). Oligo sequences can be found in Supplementary Table 3. PCR amplification of the oligo pool prior to cloning was done according to the manufacturer's protocol. For cloning the oligo pool into the appropriate lentiviral backbone the following reaction was set up: 5 μl 10× Cutsmart buffer (NEB), 1 mM DTT (final), 1 mM ATP (final), 1.5 μl T4 DNA Ligase (8000U, NEB), 1.5 μl EspI (NEB), 100 ng oligo pool PCR product and 500 ng vector (LentiGUIDE-puro, EspI-digested, and isopropanol purified) and water up to 50 μl. The reaction was incubated for 100 cycles at 5 min 37 °C followed by 5 min 20 °C. After isopropanol clean-up, the ligation was transformed into NEB Stable Competent E. coli (C3040I)

and streaked out onto LB agar plates. Library integrity was confirmed using Illumina sequencing.

**In vivo sublibrary screen.** KPC-Cas9-OVA cells were transduced with the lentiviral sublibrary at a MOI of 0.3 and selected for four days with 2 μg/ml Puromycin. Subsequently, 150.000 KPC-Cas9-OVA-Sublibrary cells were orthotopically transplanted into Rag1−/− mice. On day 16 post transplantation 1 × 10^6 preactivated OT-I CD8+ T-cells were adoptively transferred (intravenously) into tumor-bearing mice. Mice were sacrificed on day 21 post transplantation and tumors were harvested. Tumor DNA was isolated using the Qiagen Blood and Tissue Kit and sgRNA cassette was amplified similarly to the in vitro screen and analyzed by Illumina sequencing. Single guide RNA representation was assessed using MAGeCK (v0.5.6) by comparison to the plasmid sublibrary. The screening data set can be found in Supplementary Table 3 and via the GEO accession number GSE180834.

**Transplantations.** Mice were anesthetized using isoflurane at a constant flow rate. The abdomen was shaved and sterilized before a small incision in the upper left quadrant was made. The pancreas was carefully put onto a cotton-swab and 1.5 × 10^5 KPC cells were injected in 50 μl of PBS:Matrigel (1:1) using a 29 G needle. Successful injection was confirmed when a liquid bled formed and no leakage could be observed. Peritoneum and skin were subsequently sutured with Vicryl violet sutures (N385H, Ethicon) and secured with wound clips (FST). Approximately three weeks post transplantations animals were sacrificed and tumors were isolated, weighed, and processed for subsequent analysis. For subcutaneous transplantations, C57BL/6 mice were injected with 1 × 10^6 KPC or KPC-Cas9 cells per flank mix in 1:1 PBS:Matrigel. Tumors were measured with calipers and the volume was estimated via the equation $(L \times W^2)/2$. Maximal tumor burden as permitted by your ethics committee did never exceed a diameter larger than 12 mm s.c., or more than 7 mm orthotopically.

Immune check point inhibition: Animals were randomly allocated to the treatment group and received either PBS or 200 μg/mouse anti-PD-1 (BioXcell invivomab BE0146) and 200 μg/mouse anti-CTLA4 (BioXcell invivomab BE0131) every third day starting at day 7 post transplantation (total of five injections maximal), followed by a three-day wash-out phase before tumor analysis.

**Flow cytometry for tumor microenvironment analysis.** For flow cytometry analysis orthotopic tumors were collected and minced into small pieces before digestion in Collagense IV (6000 U/ml) and DNase I (200 U/ml) for one hour at 37 °C. Cell suspension was filtered through a 40 μm cell strainer. For cytokine stainings, cells were restimulated with PMA (100 nM), ionomycin (1 μg/ml), and monensin (2 μg/ml) for 3–4 h at 37 °C in complete IMDM medium (Gibco). Viability staining was performed using the fixable viability dye eFluor780. Staining with fluorescent antibodies was carried out for 15 min at 4 °C in the dark. After washing in FACS buffer (2 mM EDTA, 2% FBS), cell suspension was acquired using BD Fortessa and FlowJo software (Treestar).

**Antibodies used for flow cytometry.** PD-1 FITC (1:300; J43; eBioscience), NK1.1 PE (1:300; PK136; eBioscience), CD3e PE-Dazzle594 (1:300; 145-2C11; BioLegend), CD3e PE (1:200; 145-2C11; eBioscience), FoxP3 PerCP-Cy5.5 (1:200; FJK-16s; eBioscience), CD8a PE-Cy7 (1:300; 53-6.7; eBioscience), CD8a APC (1:600; 53-6.7; eBioscience), CD8a PerCP-Cy5.5 (1:400; 53-6.7; eBioscience), TCRb AF700 (1:100; H57-597; BioLegend), CD45 BV785 (1:1000; 30-F11; BioLegend), CD4 BV711 (1:400; GK1.5; BioLegend), CD19 BV650 (1:400; 6D5; BioLegend), CD19 PE (1:500; 1D3; eBioscience), CD11b BV605 (1:1000; M1/70; BioLegend), CD11b BV510 (1:400; M1/70; BioLegend), CD11c BV605 (1:400; N418; BioLegend), Siglec-F PE (1:300; E50-2440; BD Biosciences), F4/80 APC (1:200; BM8; BioLegend), Ly-6G AF 700 (1:300; 1A8; BioLegend), MHC II BV650 (1:4000; M5/114.15.2; BioLegend), CD64 BV421 (1:200; X54-5/7.1; BioLegend), TNF FITC (1:300; MP6-XT22; BioLegend), IFNg PE-Cy7 (1: 1000; XMG1.2; BD Biosciences), Eomes PE (1:200; W17001A; BioLegend), Ki-67 BV 650 (1:200, 11F6; Biolegend), GzmB FITC (1:100; GB11; BioLegend). Gating strategy is depicted in Supplementary Fig. 6d.

**RNA-Seq.** After 6 h of co-culture, OT-I T-cells and KPC-Cas9-OVA cancer cells were sorted using the BD Aria cell sorter. KPC cells expressed mCherry, T-cells were stained with CD8a-APC (clone 53-6.7; Biolegend 100711; 1:600) to separate both populations. RNA was isolated with the Qiagen RNeasy Mini Kit and sent to the Functional Genomic Center Zurich (FGCZ) for standard library preparation and Illumina sequencing. Read quality was checked using FastQC. Reads were trimmed using cutadapt and mapped to the mouse genome GRCm38 with HISAT2 followed by sorting using samtools. The raw count matrix was generated in RStudio using Rsubread. Differential gene (DE) expression analysis was performed with EdgeR and differentially expressed genes (LFC ± 1; FDR < 0.1) were used as input for GO term analysis using the Molecular Signature Database (MSigDB). RNA-Seq data can be accessed via GEO.

**TNF treatment of KPC cells.** KPC cells were treated 24 h with the indicated TNF concentration and stained subsequently for immunofluorescence (see below) or

with crystal violet to assess cell viability (here in the presence of 1 µg/ml Actinomycin D). Crystal violet dye was reconstituted in 10% acetic acid and absorbance was measured at 595 nm.

**Immunofluorescence of KPC cells**. Cells were grown on glass cover slips, treated with TNF (100 ng/ml), and fixed for 10 min at room temperature in 4% PFA. Cells were permeabilized and blocked in 0.5% Triton-X, 5% normal donkey serum in PBS. Cleaved caspase 3 antibody (1:400, Cell signaling Technology, #9664) was diluted in blocking solution and incubated overnight at 4 ˚C. Coverslips were washed in PBS and incubated 2 h at room temperature with secondary antibody (Donkey anti-rabbit-568, ThermoFisher Scientific, 1:400), Alexa Fluor 647 Phalloidin (1:1000, Cell signaling technology, #8940) and DAPI. Coverslips were mounted with Prolong Gold (ThermoFisher Scientific) and imaged with a Lunaphore.

**Whole-mount staining of human pancreatic cancer organoids**. Wildtype or $RNF31^{KO}$ hPan/hPDA organoids were treated for 4 h with 100 ng/ml human TNF (Peprotech) in 8-well µ-slides (Ibidi). After fixation in 4% PFA, organoids were blocked and permeabilized in blocking solution (10% normal donkey serum; 0.5% Triton-X in PBS). All antibody incubations were performed overnight at 4 ˚C on a rocking platform. Primary antibodies: E-Cadherin (1:500, R&D Systems, AF748), cleaved Caspase 3 (1:400, Cell Signaling Technology, 9664). Donkey-anti-goat 488 and donkey-anti-rabbit 568 were used as secondary antibodies and counterstained with DAPI (always 1:400 for secondary antibodies). Organoids were mounted with ProLong Gold. Confocal Images were taken with a Zeiss LSM 880 Airyscan. Organoids were quantified manually using Fiji Image J by counting three individual images from independent experiments per condition.

**Western blot**. Whole-cell lysates were prepared in RIPA buffer (50mMTris-HCl pH 8.0, 150mMNaCl, 0.1% SDS, 0.5% Na-Deoxycholate, 1%IGEPAL CA-630) supplemented with PhosSTOP phosphatase inhibitors and cOmplete protease inhibitor cocktail (both Roche). BCA protein assay (ThermoScientific) was used for protein quantification. Samples were loaded on 4–15% precast polyacrylamide gels (Bio-Rad) and transferred to PVDF (Bio-Rad) membranes in Towbin buffer. Membranes were blocked in 5% bovine serum albumin (Applichem) and incubated overnight in primary antibodies phospho-p65 (1:1000; CST#3033), Caspase 8 (1:1000; CST#4790), cleaved Caspase 8 (1:1000, CST#8592), FLIP (1:1000, CST#56343) and Gapdh (1:3000; CST#14C10). IRDye800CW and 680RD donkey anti-rabbit secondary antibodies were used for detection at a 1:10,000 dilution (LI-COR). Protein bands were visualized with the ODYSSEY CLx imaging system (LI-COR).

**Genomic DNA isolation and next-generation sequencing**. Genomic DNA from murine KPC cells war extracted using direct cell lysis buffer: 10 µl of 4× lysis buffer (10 mM Tris-HCl pH 8, 2% Triton X-100, 1 mM EDTA, 1% freshly added proteinase K) and incubated at 60 °C for one hour and inactivated at 95 °C for 10 min. Target sites of sgRNAs in Vps4b or Rnf31 were amplified by PCR using GoTaq G2 Hot Start Green Master Mix (Promega) and NEBNext High-Fidelity 2× PCR Master Mix for adding i501 and i701 Illumina sequencing adapters in a second PCR with 8 cycles. PCRs were purified with 0.8× Agencourt AMPure XP beads (Beckman Coulter) and measured using the Qubit 3.0 fluorometer. Samples were sequenced on Illumina MiSeq and analyzed using CRISPResso2.

**Primers used to amplify target loci**. Rnf31-1 + 2: FW 5'CTTTCCCTACACG ACGCTCTTCCGATCTNNNNNNNGGGCTAGGCTCACGACTCTGCT
Rnf31-1 + 2: RV 5′ GGAGTTCAGACGTGTGCTCTTCCGATCTNNNNNNN CAGCAAACTGAGCTCGGGTGCGA
Rnf31-3: FW 5′ CTTTCCCTACACGACGCTCTTCCGATCTNNNNNNTC CTCCTGAATCACATTGCAG
Rnf31-3: RV 5′ GGAGTTCAGACGTGTGCTCTTCCGATCTNNNN NNCACTAGGTAGGCAGAGGCTTAC
Vps4b-1: FW 5′ CTTTCCCTACACGACGCTCTTCCGATCTNNNNN NCCACCCACTGTAGACATT
Vps4b-1: RV 5′ GGAGTTCAGACGTGTGCTCTTCCGATCTNNNNNNNC CATGGTTAGAAAATCCCTG
Vps4b-2: FW 5′ CTTTCCCTACACGACGCTCTTCCGATCTNNNNNN GGGGCTTTAGTGCCGTTCAGG
Vps4b-2: RV 5′ GGAGTTCAGACGTGTGCTCTTCCGATCTNNNN NNTGGAAAGTGTCTTGCAATCCAACA
Vps4b-3: FW 5′ CTTTCCCTACACGACGCTCTTCCGATCTNNNNN NTCAGTGTGGCTAATGTCTGAAGT
Vps4b-3: RV 5′ GGAGTTCAGACGTGTGCTCTTCCGATCTNNNNNNNTTT CGCTTTCCCCCAGCCACTT

**Human expression data**. Human expression data were retrieved from CTGA and GTEx data bases and visualized using the online platform GEPIA2 (http://gepia2.cancer-pku.cn/#index).

**Statistics**. To compare data from experiments we either applied student's unpaired, two-tailed T-test or One-Way ANOVA analysis, as indicated in the respective figure legend. A minimum of three independent biological replicates was performed per experiment. P values larger than 0.05 were considered as non-significant. Exact p values are displayed in the respective panels. Statistical comparisons were performed using RStudio and GraphPad Prism 9.

**Reporting summary**. Further information on research design is available in the Nature Research Reporting Summary linked to this article.

## Data availability

RNA-Seq data and all CRISPR screening data have been made accessible via GEO: GSE180834. Mouse genome GRCm38 was used as reference. Human patient data in Supplementary Fig. 2b can be accessed via GEPIA2 [http://gepia2.cancer-pku.cn/#index]. The remaining data are available within the Article, Supplementary Information or Source Data file. Source data are provided with this paper.

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

## Acknowledgements

We thank the Flow Cytometry Facilities of the University of Zurich and the ETH Zurich. Also, we thank the Functional Genomics Center Zurich and the ETH Phenomics Center for their support and infrastructure. We also thank Chantal Pauli and Daniela Lenggenhager as well as the University Hospital Zurich Organoid Biobank for their help in providing patient samples. This work was supported by the Swiss National Science Foundation grant 310030_185293 (G.S.), the Swiss National Science Foundation grant 310030B_182829 (M.K.), ETH PhD Fellowship (N.F.), PHRT iDoc Fellowship PHRT_324 (K.M.), and EMBO Long-Term 499 Fellowship ALTF 873-2019 (S.J.).

## Author contributions

N.F. and G.S. conceptualized the study. N.F. performed experiments, analyzed the data, and wrote the manuscript. L.T. designed, supervised, and helped analyzing flow cytometry experiments and gave valuable input throughout the course of the project. S.J. helped analyzing RNA-Seq data. D.E., K.F.M., T.R., and F.A. performed experiments. N.F. and G.S. wrote the manuscript, L.T. and M.K. reviewed and edited the manuscript. G.S. supervised the study. G.S. and M.K. acquired funding. All authors approved the final version of the manuscript.

## Competing interests

The authors declare no competing interests.
