## [Peer Review File · Nature Communications]

Reviewers' Comments:

Reviewer #1:

Remarks to the Author:

Frey et al describe present work demonstrating that loss of Rnf31 or Vps4b sensitize mouse pancreatic cancer cells to T-cell mediated killing. They performed an in vitro genome-scale CRISPR screen in the presence of CD8+ T-cells and looked for knockout events that sensitize or cause resistance to CD8-mediated T-cell killing. They then followed up with an in vivo screen with a targeted library. Findings converged on several known findings as well as the identification of Rnf31 and Vps4b as two potentially novel sensitizers to immune cell mediated killing. The authors went on to validate Rnf31 and Vps4b as sensitizers in both in vitro and in vivo assays, including competition assays. They provide preliminary mechanistic data to suggest that Rnf31 knockout causes increased sensitivity to TNF induced apoptosis through Caspase 8 in a manner that does not depend on adaptive immunity. Vps4b knockout impairs autophagy leading to insufficient clearance of granzyme B derived from CD8+ T-cells. This phenotype requires adaptive immunity. The screen and validation data are interesting and Rnf31 and Vps4b appear to be novel sensitizers to immune-mediated killing. The manuscript could be improved in several ways.

The authors validate Rnf31 and Vps4b in primarily KPC cell lines and it is unclear how many sgRNA reagents and how many different lines they have used. These experiments should be done with at least two independent sgRNA reagents, demonstrating sufficient knockout by immunoblot, and then should be employed in multiple different KPC lines. Additionally, it would be important to also investigate these sensitizing factors in human organoid + T-cell co-culture systems, outside of the Ova+ KPC cell lines system.

The authors do not clearly show the biochemical mechanism by which Rnf31 loss leads to sensitization to TNF. Further work in this regard would improve the impact of this paper.

Vps4b has a paralog protein, Vps4a, that plays a functionally redundant role. Why does Vps4a not score in the screen? What about other members of the ESCRT pathway. The authors should specifically investigate Vps4a in their in vitro and in vivo validation assays.

Does Vps4b knockout impair autophagy and lead to reduced granzyme B clearance in human PDAC cells exposed to CTLs?

Human organoid work in Figure 4 is shown qualitatively but these results should be better quantitated and presented to the reader.

The in vivo experiments in figure 6 should be done as a tumor maintenance experiment, knocking down or knocking out the gene in the context of an established tumor. The in vivo experiments also need additional labeling – how many mice were examined?

Reviewer #2:

Remarks to the Author:

Immunotherapies for pancreatic cancer have been failing to benefit patients in clinical trials. Identifying factors to sensitize pancreatic cancer cells to T cell killing could offer a new avenue for therapeutic strategies in clinic. Frey et al. present an interesting study that utilized a genome-scale CRISPR screen approach in an two cell-type (2CT) assay and in vivo model of pancreatic ductal adenocarcinoma (PDA) to identify cell-intrinsic mechanisms of resistance to cytotoxic T cells (CTL) killing. Enrichment of IFN γ signaling and antigen presentation genes in the CRISPR screens suggests the screening approach and controls were appropriate. Here, authors identified novel role of Rnf31 and Vps4b as essential factors required for the escape of PDA cells from CD8 T cell killing and its underlying mechanisms. Hence, the data presented here is an important advancement in the field in terms of methodology and relevance of the findings. However, the study has certain fundamental issues that need to be addressed to defend and substantiate the claims made here.

Comments:

The authors say that loss of Rnf31 and Vps4b results in enhanced CTL killing throughout the entire

manuscript, however they do not report any evidence confirming that Rnf31 and Vps4b were deleted in KPC and human PDA organoid cells. The genetic knockout in the cells must be validated at either DNA, transcriptional or protein levels. Without the confirmation of gene knockouts, it would be difficult to consider all the claims made here valid.

The lentiCRISPR system has been used here to introduce gene deletions in PDA cells. The constitutive expression of sgRNA and Cas9 results in off-target indel mutations which makes it difficult to assess whether the phenotypic changes observed in the cells are due the deletion of an intended gene or due to unwanted genomic aberrations. This can be addressed by: 1. Performing DNA sequencing of the cell-lines to confirm that there were no off-target deletions e.g qEva-CRISPR method reported by Dabrowska et al. *Nucleic Acids Research* 2018; Or 2. Confirming that Rnf31 and Vsp4b gene deletion with 2 or more sgRNAs give the same phenotypic effect of the PDA sensitizing to CTL killing.

Authors should provide a clear rationale and explanation on why Rnf31 and Vsp4b were selected for mechanistic studies out of dozens of other validated CRISPR screen hits.

In lines 136-140, the authors generated the hypothesis Cxcr3 ligands may be regulated by Rnf31 and Vps4b based on gene expression data, which has not been followed up in the paper. To improve the relevancy of this finding, can this observation be associated with human tumors? Can loss of Rnf31 or Vps4b be associated with the expression of Cxcr3 ligands and T cell infiltrations in human tumors using public datasets?

Rnf31 is linked with the p53 pathway and autophagy (Chu et al. *Autophagy* 2021; Zhu et al. *Oncogene* 2016). It will be useful to test if p53 is degraded rapidly upon Rnf31 deletion in PDA cells and if this mechanism synergizes with the reported TNF mechanism.

Supplementary Figure 4C shows that cFLIP destabilization is the key link between the mechanism of Rnf31 loss, phospho-p65 and Caspase 8. Authors would need to validate this link by measuring cFLIP destabilization.

Figures 4d and Supp. 4b shows the drastic effect of an apoptosis trigger with RNF31 loss. However no quantification of the image data has been presented.

Figure 6 showed compelling evidence that introduction of sgRNAs encoding Rnf31 and Vsp4b in KPC-Cas9 cells favors T cell function in vivo through increase in CD8 T cell infiltrates and effector function. A major immunosuppressive component of human PDA is stroma. Do Rnf31 and Vsp4b gene loss sensitize the tumors via any effects on the stromal cells of PDA?

In line 230, PD1+ CD8+ T cells are referred to as exhausted cells. Are these cells really exhausted? What are the expression levels of Lag3, Tim3 and Ki67 in these cells?

Authors have used human pancreatic and PDA organoids in the study. Tissue source and IRB approvals for the use of human tissues should be mentioned in the methods.

A typo should be corrected on line 339 of the methods section.

Reviewer #3:

Remarks to the Author:

Frey et al. provide an interesting research story investigating how the Rnf31 and Vps4b mediate T cell mediated killing in pancreatic cancer. The manuscript is well written and provides clear figures throughout. Addressing some of the below comments could strengthen this work:

Line 28: Introduction is very brief and could be expanded to provide context why this study is unique. This group appears to have used this CRISPR approach in other cancers and you want to understand what makes this study novel.

Line 61-63: Provide level of cytotoxicity they observed with this in vitro killing assay. How cytotoxic are these T cells

Line 89: Why were only Rag KO mice used for these studies? Understanding development of endogenous T cell activity by injecting KPC-Cas9-OVA cells into OT-1 mice might be more relevant. Tetratmer staining and other methods could be utilized to follow level of CD8 T cell mediated endogenous response.

Line 106: Please further clarify "per-se"...there does seem to be slight decrease, but not at the final time point.

Results: What about Rnf31 and Vps4b expression in patient samples? Does low expression correlate to a change in patient survival? Providing this data or commenting in the discussion would be important for readers to know.

Results: Since Rnf31 and Vps4b sensitize PDAC to T cell mediated killing, does this result in checkpoint immunotherapy efficacy in vivo?

Discussion: Discussion needs to be more thorough and better developed. Currently it is lacking. Possibly discuss autophagy inhibitors in PDAC trials which have not shown much promise. Are expression of these genes unique to PDAC. Since these genes (Rnf31 and Vps4b) were discovered due to CTL selection in an OT-1 ova model system, is this true in PDAC where antigens might not be highly present?

We thank all three reviewers for their constructive and valuable input on our manuscript. For a detailed point-by-point response please see below. In summary, in the revised manuscript we confirm the phenotype for loss of Rnf31 and Vps4b using several individual sgRNAs per candidate gene, validate indel formation on the DNA level and tested candidate knock-outs in additional cell lines. Regarding Vps4b, we also assessed whether elimination of other ESCRT-III members sensitizes PDA cells to T cell killing and whether the observed phenotype is conserved in human cells. Regarding Rnf31, we provide additional mechanistic insights and demonstrate that loss of Rnf31 facilitates induction of caspase 8 mediated apoptosis via destabilization of c-Flip, an inhibitor of apoptosis. Finally, we improved our in vivo analysis by (i) assessing T cell exhaustion in depth using several markers and (ii) by combining Rnf31 and Vps4b elimination with immunotherapy, which further enhanced the observed phenotypes.

REVIEWER COMMENTS

Reviewer #1 (Remarks to the Author): with expertise in pancreatic cancer; CRISPR screens, organoids

Frey et al describe present work demonstrating that loss of Rnf31 or Vps4b sensitize mouse pancreatic cancer cells to T-cell mediated killing. They performed an in vitro genome-scale CRISPR screen in the presence of CD8+ T-cells and looked for knockout events that sensitize or cause resistance to CD8-mediated T-cell killing. They then followed up with an in vivo screen with a targeted library. Findings converged on several known findings as well as the identification of Rnf31 and Vps4b as two potentially novel sensitizers to immune cell mediated killing. The authors went on to validate Rnf31 and Vps4b as sensitizers in both in vitro and in vivo assays, including competition assays. They provide preliminary mechanistic data to suggest that Rnf31 knockout causes increased sensitivity to TNF induced apoptosis through Caspase 8 in a manner that does not depend on adaptive immunity. Vps4b knockout impairs autophagy leading to insufficient clearance of granzyme B derived from CD8+ T-cells. This phenotype requires adaptive immunity. The screen and validation data are interesting and Rnf31 and Vps4b appear to be novel sensitizers to immune-mediated killing. The manuscript could be improved in several ways.

The authors validate Rnf31 and Vps4b in primarily KPC cell lines and it is unclear how many sgRNA reagents and how many different lines they have used. These experiments should be done with at least two independent sgRNA reagents, demonstrating sufficient knockout by immunoblot, and then should be employed in multiple different KPC lines.

We thank the reviewer for this remark and performed additional experiments to address this point. We now applied 3 individual sgRNAs for Rnf31 and Vps4b, which all led to the same phenotype in the competition assay, respectively (Figure 3c). In Supplementary Figure 3a we show deep sequencing of the genomic site of every sgRNA and confirm an average indel formation rate of over 80% for each sgRNA, and in Figure 3d we confirm that Rnf31 and Vps4b also sensitize tumor cells to CTLs in an additional KPC cell line.

Additionally, it would be important to also investigate these sensitizing factors in human organoid + T-cell co-culture systems, outside of the Ova+ KPC cell lines system.

While we agree with the reviewer that this would be an interesting experiment, we would like to point out that we do not have PDAC organoid lines with matched T cells from blood samples in our lab, and that it would be extremely difficult and time consuming to generate such lines. First, we currently do not have a clinical partner for obtaining PDAC biopsies, and second we do not have an ethical approval for isolating organoid lines together with peripheral blood. From personal communication with

researches that have recently generated a biobank of colorectal cancer organoids together with matched T cells, we also know that this is a very tedious process: in their hands they were only successful in establishing matched organoids and T cells in 10% of patients. Thus, establishing the required lines would certainly take > 1 year, and would therefore be beyond the scope of a revision. Theoretically, the NY-ESO-1 system could theoretically be applied to organoids as an alternative to matched organoid T cell pairs. However, when expressing the NY-ESO-1 antigen and transducing human T cells with the respective TCR, one would need to obtain organoids from HLA-A*02:01 donors to ensure NY-ESO-1 surface presentation, which would again require establishing a novel biobank.

The authors do not clearly show the biochemical mechanism by which Rnf31 loss leads to sensitization to TNF. Further work in this regard would improve the impact of this paper.

We fully agree with the reviewer's point and conducted additional experiments to elucidate the underlying mechanisms of how Rnf31 loss leads to increased TNF sensitivity. First, we performed Western Blot analysis (Figure 4c) of the anti-apoptotic protein c-Flip that is thought to be stabilized by Rnf31-mediated ubiquitination. We found that c-Flip was degraded in the absence of Rnf31 upon TNF treatment, suggesting that Rnf31 is indeed essential for stabilizing this inhibitor of apoptosis. We also tested whether loss of *Cflar* (gene encoding for c-Flip) mimics our phenotype observed in Rnf31 knockout cells. Indeed, in *Cflar*^{KO} KPC cells we observed immediate induction of apoptosis upon TNF treatment (Supplementary Figure 4d), mimicking the phenotype of Rnf31^{KO} KPC cells. Altogether, these data suggest that Rnf31 prevents TNF-mediated apoptosis via stabilizing c-Flip.

Vps4b has a paralog protein, Vps4a, that plays a functionally redundant role. Why does Vps4a not score in the screen? What about other members of the ESCRT pathway. The authors should specifically investigate Vps4a in their in vitro and in vivo validation assays.

Indeed, our screen only identified *Vps4b* as a candidate to sensitize cancer cells to T cell immunity upon depletion, and the isoform Vps4a as well as other members of the ESCRT complex did not score as hits. Interestingly, the same was observed in genome wide CRISPR screens for T cell immunity in melanoma, colon, breast and kidney cancer cells (Pan et al; Science 2018; Lawson et al; Nature 2020), already indicating that the role in T cell mediated killing of tumor cells is specific to Vps4b. When we assessed if loss in Vps4a, Vta1 (a co-factor for Vps4a/b ATPase activity) or the scaffolding protein Chmp4b lead to similar effects as loss of Vps4b, we indeed found that inhibiting these ESCRT-III complex components did not enhance T cell mediated killing of KPC cells. We added these data to the revised manuscript (Supplementary Figure 3e).

Does Vps4b knockout impair autophagy and lead to reduced granzyme B clearance in human PDAC cells exposed to CTLs?

We appreciate the reviewer's point and addressed it by applying our autophagy reporter in human cells. When we tested the autophagy flux probe in commonly used Panc-1 cells, we found that the reporter itself was not responding to starvation as observed in murine KPC cells. We assume that this could be due to the fact that Panc-1 cells are tumor-derived and could therefore already have a modified autophagy regulation. We therefore decided to use HEK293T cells for the assay (Mizushima Lab, Tokyo). While we found that loss in *ATG9A* strongly impedes autophagic flux, loss of *VPS4B* did not impair autophagy regulation in human cells (Supplementary Fig. 5c). Thus, it seems that the Vps4b knock out phenotype is not conserved between mouse to human cells (also described in the discussion of the revised manuscript - line 290).

Human organoid work in Figure 4 is shown qualitatively but these results should be better quantitated and presented to the reader.

We thank the reviewer for this important remark and we now include a quantification of the depicted images in Figures 4e/f as well as Supplementary Figures 4b/c.

The in vivo experiments in figure 6 should be done as a tumor maintenance experiment, knocking down or knocking out the gene in the context of an established tumor. The in vivo experiments also need additional labeling – how many mice were examined?

We agree that a tumor maintenance experiment would give valuable insights to whether Rnf31/Vps4b could serve as drug targets, as chemical compounds inhibiting these factors would be administered to patients with fully grown tumors. However, to our knowledge it is not feasible to efficiently deliver CRISPR-Cas9 components into established tumors, as common vectors such as AAVs, lipid nanoparticles (LNPs) or Adeno-viruses (AdVs) do not efficiently transduce tumor cells in vivo. In fact, we had previously tested RFP and GFP delivery into the pancreas or transplanted KPC cells using systemic delivery of these three vectors, but did not obtain relevant expression. As we would require transduction rates (and subsequent knockout) of over 80% to draw a conclusion whether Vps4b or Rnf31 loss could be exploited for tumor therapy or not, we believe the experiment is not feasible with currently available delivery vectors. Another option would be the use of inducible Cas9 systems, but in our hands, these are inherently leaky and also do not allow the induction of a gene knock-out at a later timepoint in vivo.

Reviewer #2 (Remarks to the Author): with expertise in cancer immunology, CRISPR screen

Immunotherapies for pancreatic cancer have been failing to benefit patients in clinical trials. Identifying factors to sensitize pancreatic cancer cells to T cell killing could offer a new avenue for therapeutic strategies in clinic. Frey et al. present an interesting study that utilized a genome-scale CRISPR screen approach in an two cell-type (2CT) assay and in vivo model of pancreatic ductal adenocarcinoma (PDA) to identify cell-intrinsic mechanisms of resistance to cytotoxic T cells (CTL) killing. Enrichment of IFN γ signaling and antigen presentation genes in the CRISPR screens suggests the screening approach and controls were appropriate. Here, authors identified novel role of Rnf31 and Vps4b as essential factors required for the escape of PDA cells from CD8 T cell killing and its underlying mechanisms. Hence, the data presented here is an important advancement in the field in terms of methodology and relevance of the findings. However, the study has certain fundamental issues that need to be addressed to defend and substantiate the claims made here.

Comments:

The authors say that loss of Rnf31 and Vps4b results in enhanced CTL killing throughout the entire manuscript, however they do not report any evidence confirming that Rnf31 and Vps4b were deleted in KPC and human PDA organoid cells. The genetic knockout in the cells must be validated at either DNA, transcriptional or protein levels. Without the confirmation of gene knockouts, it would be difficult to consider all the claims made here valid. The lentiCRISPR system has been used here to introduce gene deletions in PDA cells. The constitutive expression of sgRNA and Cas9 results in off-target indel mutations which makes it difficult to assess whether the phenotypic changes observed in the cells are due the deletion of an intended gene or due to unwanted genomic aberrations. This can be addressed by: 1. Performing DNA sequencing of the cell-lines to confirm that there were no off-target deletions e.g qEva-CRISPR method reported by Dabrowska et al. Nucleic Acids Research 2018; Or 2. Confirming that Rnf31 and Vsp4b gene deletion with 2 or more sgRNAs give the same phenotypic effect of the PDA sensitizing to CTL killing.

We thank the reviewer for raising this important point. We improved the manuscript in several ways: 1. We show in Figure 3c three individual sgRNAs for Rnf31 and Vps4b, all exhibiting the same phenotype in the competition assay. 2. We validated the knock-out efficiency of the different sgRNAs by deep sequencing each target site individually and found an insertion/deletion frequency of more than 80% across all different sgRNAs (Supplementary Figure 3a). As we observe the same phenotype with all sgRNAs per gene, we are confident that our observed effect is not due to off-targets of one

sgRNA. Furthermore, in Supplementary Figure 3d we show that RNA transcripts of candidates are downregulated in our RNA-Seq experiment where we used a pool of all three sgRNAs per gene.

Authors should provide a clear rationale and explanation on why Rnf31 and Vsp4b were selected for mechanistic studies out of dozens of other validated CRISPR screen hits.

We agree with the reviewer that a rationale may help to understand our focus on the role of Rnf31 and Vps4b. In lines 41-44 as well as in lines 104-107 we strengthen this point. In short, we were mainly interested in exploring the mechanism of those two genes, as to our knowledge they have not been described to increase CTL-sensitivity in PDA or any other cancer type before. Moreover, little has been known about a potential role of Vps4b in cancer.

In lines 136-140, the authors generated the hypothesis Cxcr3 ligands may be regulated by Rnf31 and Vps4b based on gene expression data, which has not been followed up in the paper. To improve the relevancy of this finding, can this observation be associated with human tumors? Can loss of Rnf31 or Vps4b be associated with the expression of Cxcr3 ligands and T cell infiltrations in human tumors using public datasets?

We checked human data sets (TCGA) for correlation of high/low expression of RNF31/VPS4B with CXCR3 chemokines. However, as we did not find a significant correlation, we adjusted the passage in the manuscript accordingly to weaken the hypothesis. See lines 157 – 159.

Rnf31 is linked with the p53 pathway and autophagy (Chu et al. Autophagy 2021; Zhu et al. Oncogene 2016). It will be useful to test if p53 is degraded rapidly upon Rnf31 deletion in PDA cells and if this mechanism synergizes with the reported TNF mechanism.

As p53 is mutated (non-functional) in KPC cells we do not think that p53 degradation synergizes with the observed sensitivity to TNF in Rnf31^{KO} cells.

Supplementary Figure 4C shows that cFLIP destabilization is the key link between the mechanism of Rnf31 loss, phospho-p65 and Caspase 8. Authors would need to validate this link by measuring cFLIP destabilization.

We fully agree with the reviewer's point and performed Western Blot analysis for c-Flip at different timepoints after TNF exposure in WT and Rnf31^{KO} KPC cells (Figure 4c). We found that c-Flip is degraded during TNF treatment in the absence of Rnf31. We furthermore show that genetic disruption of Cflar (gene encoding for c-Flip) mimics the phenotype observed in Rnf31-KO cells (Supplementary Figure 4d). In line with these findings, *Cflar* has also been depleted in our genome-wide CRISPR screen in vitro and targeted CRISPR screen in vivo. Together these data suggest that Rnf31 acts upstream of c-Flip (*Cflar*) to regulate the response to TNF.

Figures 4d and Supp. 4b shows the drastic effect of an apoptosis trigger with RNF31 loss. However no quantification of the image data has been presented.

In the revised manuscript we adapted Figure 4e and Supplementary Figure 4b and included quantifications for apoptotic cells.

Figure 6 showed compelling evidence that introduction of sgRNAs encoding Rnf31 and Vsp4b in KPC-Cas9 cells favors T cell function in vivo through increase in CD8 T cell infiltrates and effector function. A major immunosuppressive component of human PDA is stroma. Do Rnf31 and Vsp4b gene loss sensitize the tumors via any effects on the stromal cells of PDA?

We thank the reviewer for this comment. Indeed, the immunosuppressive stroma has been recognized as a critical mediator of disease progression in PDA. However, our study mainly relied on an orthotopic transplantation model that allowed us to test different genotypes in vivo in a timely manner. The downside of this tumor model is the insufficient stroma formation as tumors grow rapidly and the majority of the tumor bulk is therefore composed of tumor cells. Hence, our model did not

allow us to draw meaningful conclusions on alterations of the TME. We discuss this limitation of our study in the revised manuscript (lines 305-310).

In line 230, PD1+ CD8+ T cells are referred to as exhausted cells. Are these cells really exhausted? What are the expression levels of Lag3, Tim3 and Ki67 in these cells?

When we analyzed Lag3 and Tim3 in tumor infiltrating lymphocytes across all genetic backgrounds we did not observe any surface expression, while PD-1 was expressed (see Figure 6c in the revised manuscript). PD-1 was strongly reduced in Vps4b and Rnf31 knockouts, concomitant with an increase in effector cytokines. When we looked at proliferation, we observed reduced Ki67+ CD8+ T cells in candidate KO tumors. We suspect that this phenotype was caused by the reduced tumor mass at the time point of analysis in Vps4b- and Rnf31- KO cancers, so that there was less antigen present in the tumor to stimulate T cell proliferation. The higher number of CD8+ T cells in Rnf31/Vps4b-KO settings may be explained by increased proliferation of CD8 T cells at an earlier time point that subsequently led to tumor shrinkage.

Authors have used human pancreatic and PDA organoids in the study. Tissue source and IRB approvals for the use of human tissues should be mentioned in the methods. We adapted the methods section accordingly (lines 462 – 466).

A typo should be corrected on line 339 of the methods section. We corrected the typo in the revised manuscript.

Reviewer #3 (Remarks to the Author): with expertise in pancreatic cancer, cancer immunology,

Frey et al. provide an interesting research story investigating how the Rnf31 and Vps4b mediate T cell mediated killing in pancreatic cancer. The manuscript is well written and provides clear figures throughout. Addressing some of the below comments could strengthen this work:

Line 28: Introduction is very brief and could be expanded to provide context why this study is unique. This group appears to have used this CRISPR approach in other cancers and you want to understand what makes this study novel.

We thank the reviewer for this suggestion and extended the introduction accordingly (lines 41-44 and 104-107).

Line 61-63: Provide level of cytotoxicity they observed with this in vitro killing assay. How cytotoxic are these T cells

We always used an effector to target ratio of 1:1. For method details see line 376 (genome-wide screen) and line 415 (competition assay) in the revised manuscript. In the genome-wide screen the conditions we used killed about 60 – 70% of KPC cells. We updated this in the manuscript (line 69). This was the desired killing efficiency determined with a titration assay. In the figure below, one can see the dose-dependency of OT-I T cells regarding their killing capacity. T cells and KPC cells were co-cultured for 2d in this experiment.

Line 89: Why were only Rag KO mice used for these studies? Understanding development of endogenous T cell activity by injecting KPC-Cas9-OVA cells into OT-1 mice might be more relevant. Unfortunately, OVA expressing KPC cells were rejected by C57BL/6 hosts. Therefore, we performed the OVA-OT-I screen in RAG1-deficient hosts. Nevertheless, we agree with the reviewer that endogenous T cell activity is of great interest, and therefore performed all candidate validation experiments in immunocompetent hosts (Figure 6).

Tetramer staining and other methods could be utilized to follow level of CD8 T cell mediated endogenous response.

As we were not able to transplant OVA-expressing tumors into immunocompetent mice (rejection of cells by the host), we did not know any KPC-specific antigen that could be used to follow T cell responses by using tetramer. However, measurement of effector cytokine responses (i.e. IFN γ and TNF α) produced by T cells in the tumor are likely coming from tumor-specific T cells.

Line 106: Please further clarify “per-se”...there does seem to be slight decrease, but not at the final time point.

We agree with the reviewer and adapted this sentence in the manuscript (line 115). Throughout the course of our experiments, we never observed growth differences across different KPC-knockout lines.

Results: What about Rnf31 and Vps4b expression in patient samples? Does low expression correlate to a change in patient survival? Providing this data or commenting in the discussion would be important for readers to know.

We included patient expression data in Supplementary Figure 2b. For VPS4B and RNF31 we found that there is indeed an upregulation compared to normal tumor tissue, indicating that expression of either gene is of advantage for tumor cells. This in turn supports our findings, that loss of RNF31 or VPS4B could potentially be interesting to target.

Results: Since Rnf31 and Vps4b sensitize PDAC to T cell mediated killing, does this result in checkpoint immunotherapy efficacy in vivo?

We thank the reviewer for this valuable input and in the revised manuscript we included results from an in vivo experiment where we compared Rnf31-KO (or Vps4b-KO) tumors with or without immune checkpoint blockade (ICB; anti-PD1 and anti-CTLA4) (Figure 6d). Importantly, immunotherapy led to an additional increase in CD8⁺ T cell infiltration in Rnf31-KO or Vps4b-KO cells, and to a reduction in surface PD-1 expression. Effector cytokines were increased in Rnf31-KO tumor after ICB treatment, but not in Vps4b-KO tumors as loss of Vps4b alone already substantially increase cytokine expression in T cells. Moreover, upon ICB treatment tumor mass was also further reduced in Rnf31-KO tumors.

Discussion: Discussion needs to be more thorough and better developed. Currently it is lacking. Possibly discuss autophagy inhibitors in PDAC trials which have not shown much promise. Are expression of these genes unique to PDAC. Since these genes (Rnf31 and Vps4b) were discovered due

to CTL selection in an OT-1 ova model system, is this true in PDAC where antigens might not be highly present?

In the revised manuscript we included a section in the discussion on autophagy inhibitors in clinical trials, and on the low antigenicity in PDAC (line 292-304). We hypothesize that Rnf31 inhibition is also effective in tumors with few antigens, as TNF can also be secreted by innate cytotoxic cells. In line with this hypothesis, our experiments in Rag1-deficient mice showed that in this background Vps4b-KO tumors grew at the same speed as the parental KPC line, while Rnf31-KO tumors were still significantly smaller (Supplementary Figure 6b).

Reviewers' Comments:

Reviewer #1:

Remarks to the Author:

The authors have sufficiently addressed my questions and concerns in a balanced manner. The manuscript is well done overall. The finding that VPS4B did not impair autophagy regulation in human cells is noted, although this is a single cell system. The authors address this limitation of their results in the discussion. The writing has improved in the revision.

Reviewer #2:

Remarks to the Author:

Frey et al. have addressed all of my concerns. The findings in the study will be valuable to the advancement of immunotherapies for difficult-to-treat PDAC cancers, and thus I recommend publishing these findings.

Reviewer #3:

Remarks to the Author:

I have no further comments.

FINAL REVIEWERS' COMMENTS

Reviewer #1 (Remarks to the Author):

The authors have sufficiently addressed my questions and concerns in a balanced manner. The manuscript is well done overall. The finding that VPS4B did not impair autophagy regulation in human cells is noted, although this is a single cell system. The authors address this limitation of their results in the discussion. The writing has improved in the revision.

Reviewer #2 (Remarks to the Author):

Frey et al. have addressed all of my concerns. The findings in the study will be valuable to the advancement of immunotherapies for difficult-to-treat PDAC cancers, and thus I recommend publishing these findings.

Reviewer #3 (Remarks to the Author):

I have no further comments.

We thank all three reviewers for their support.